# Robust group- but limited individual-level (longitudinal) reliability and insights into cross-phases response prediction of conditioned fear

**Maren Klingelhöfer-Jens**[1]*, **Mana R Ehlers**[1], **Manuel Kuhn**[1,2], **Vincent Keyaniyan**[1], **Tina B Lonsdorf**[1]

[1]Institute for Systems Neuroscience, University Medical Center Hamburg-Eppendorf, Hamburg, Germany; [2]Department of Psychiatry, Harvard Medical School, and Center for Depression, Anxiety and Stress Research, McLean Hospital, Belmont, United States

**Abstract** Here, we follow the call to target measurement reliability as a key prerequisite for individual-level predictions in translational neuroscience by investigating (1) longitudinal reliability at the individual and (2) group level, (3) internal consistency and (4) response predictability across experimental phases. One hundred and twenty individuals performed a fear conditioning paradigm twice 6 months apart. Analyses of skin conductance responses, fear ratings and blood oxygen level dependent functional magnetic resonance imaging (BOLD fMRI) with different data transformations and included numbers of trials were conducted. While longitudinal reliability was rather limited at the individual level, it was comparatively higher for acquisition but not extinction at the group level. Internal consistency was satisfactory. Higher responding in preceding phases predicted higher responding in subsequent experimental phases at a weak to moderate level depending on data specifications. In sum, the results suggest that while individual-level predictions are meaningful for (very) short time frames, they also call for more attention to measurement properties in the field.

*For correspondence:
m.klingelhoefer-jens@uke.de

**Competing interest:** The authors declare that no competing interests exist.

## Editor's evaluation

The authors assess the psychometric properties of behavioral, psychophysiological, and brain imaging measures of fear conditioning. Six-month retest reliability was generally low, whereas internal-consistency reliability was generally high. At the group level, reliability and criterion validity were generally good. Most measurements proved sensitive to data analytical choices. Results are framed within a larger discussion of the role of measurement properties in individual difference research and clinical translation and have the potential to serve as an important building block towards improvement in both these areas.

## Introduction

The increasing incidence (e.g., *Xiong et al., 2022*) and high relapse rates (*Essau et al., 2018*; *Yonkers et al., 2003*) of anxiety-related disorders call for a better understanding of anxiety- and stress-related processes which might contribute to improving existing treatments or developing more effective interventions. In the laboratory, these processes can be studied using fear conditioning paradigms (*Dunsmoor et al., 2022*; *Fullana et al., 2020*; *Milad and Quirk, 2012*).

In differential fear conditioning protocols (see *Lonsdorf et al., 2017a*) one stimulus is repetitively paired with an aversive unconditioned stimulus (US; e.g., electrotactile stimulation), and as a consequence becomes a conditioned stimulus (CS+) while another stimulus, the CS−, is never paired with the US. After this acquisition training phase, CSs are presented without the US (extinction training) leading to a gradual waning of the conditioned response. Critically, the fear memory (CS+/US association) is not erased, but a competing inhibitory extinction memory (CS+/no US association) is assumed to be formed during extinction training (*Milad and Quirk, 2012*; *Myers and Davis, 2007*). Subsequently, return of fear (RoF) can be induced by procedural manipulations such as a time delay (spontaneous recovery), a contextual change (renewal, *Vervliet et al., 2013a*), or a (re-)presentation of an aversive event (reinstatement, *Haaker et al., 2014*). Conditioned responding can be subsequently probed in an RoF test phase during which either the absence (i.e., extinction retention) or the return of conditioned responding (i.e., RoF) can be observed (*Bouton, 2004*; *Lonsdorf et al., 2017a*).

Findings from studies employing fear conditioning paradigms hold strong potential for translating neuroscientific findings into clinical applications (*Anderson and Insel, 2006*; *Cooper et al., 2022a*; *Fullana et al., 2020*; *Milad and Quirk, 2012*). More precisely, extinction learning is assumed to be the active component of exposure-based treatment (*Graham and Milad, 2011*; *Milad and Quirk, 2012*; *Rachman, 1989*; *Vervliet et al., 2013b*) and experimental RoF manipulations have been suggested to serve as a model of clinical relapse (*Scharfenort et al., 2016*; *Vervliet et al., 2013a*). Important findings in the fear conditioning field include the deficient learning of the safety signal (CS−) during acquisition training, impaired extinction learning (*Duits et al., 2015*) and the tendency of fear generalization to innocuous stimuli (*Cooper et al., 2022a*) in patients suffering from anxiety-related disorders as compared to healthy controls.

To date, both clinical and experimental research using the fear conditioning paradigm have primarily focused on group-level, basic, general mechanisms such as the effect of experimental manipulations – which is important to investigate (*Lonsdorf and Merz, 2017b*). Successful clinical translation (e.g., 'Why do some individuals develop pathological anxiety while others do not?') and particularly treatment outcome prediction (e.g., 'Why do some patients benefit from treatment while others relapse?'), however, requires that both the experimental paradigm and the measures employed allow for individual-level predictions over and above prediction of group averages (*Fröhner et al., 2019*; *Hedge et al., 2018*; *Lonsdorf and Merz, 2017b*). A prerequisite for this is that the measures show stability within and reliable differences between individuals over time. Hence, tackling clinical questions regarding individual-level predictions of symptom development or treatment outcome requires a shift toward and a validation of research methods tailored to individual differences – such as a focus on measurement reliability (*Zuo et al., 2019*). This is a necessary prerequisite for the long-term goal of developing individualized intervention and prevention programs. This further relates to the pronounced heterogeneity in symptom manifestation among individuals diagnosed with the same disorders (e.g., post-traumatic stress disorder, PTSD, *Galatzer-Levy and Bryant, 2013b*) which cannot be captured in binary clinical diagnoses as two patients with for example a PTSD diagnosis may not share a single symptom (*Galatzer-Levy and Bryant, 2013b*).

Measurement reliability has only recently gained momentum in experimental cognitive research (*Fröhner et al., 2019*; *Hedge et al., 2018*; *Zuo et al., 2019*) and can be assessed through test–retest and longitudinal reliability (i.e., test–retest reliability over longer time intervals, typically assessed through e.g., intraclass correlation coefficients, ICCs, see *Table 1*). Importantly, longitudinal reliability (for definitions and terminology, see *Table 1*) also has implications for the precision with which associations of one variable (e.g., conditioned responding) with another (individual difference) variable can be measured because the correlation between those two variables cannot exceed the correlations within, that is the reliability, of these two variables (*Spearman, 1910*).

Yet, in fear conditioning research, surprisingly little is known about longitudinal reliability at the individual level with time intervals ranging from 9 days to 8 months in prior work (*Supplementary file 1*; *Cooper et al., 2022b*; *Fredrikson et al., 1993*; *Ridderbusch et al., 2021*; *Torrents-Rodas et al., 2014*; *Zeidan et al., 2012*). Generally (details in *Supplementary file 1*), individual-level longitudinal reliability of risk ratings, skin conductance responses (SCRs), and fear potentiated startle (FPS) was within the same range (*Cooper et al., 2022b*; *Torrents-Rodas et al., 2014*) whereas it was numerically somewhat lower for the BOLD response as compared to different rating types (*Ridderbusch et al., 2021*). Longitudinal reliability at the individual level appeared higher for acquisition training

**Table 1.** Definitions of key terms (A) and data specifications applied across analyses (B).

(A)

| Term | Definition |
|---|---|
| **Internal consistency** | In our study, internal consistency refers to the reliability of **conditioned responding within experimental phases** at both time points, respectively. It provides information on the extent to which items – or in our case – trials measure the same construct (e.g., fear acquisition). Odd and even trials were splitted (i.e., split-half method), averaged per subject and correlated across the sample. |
| **Longitudinal reliability at the individual level** | Longitudinal reliability at the individual level indicates to which extent **responses within the same individuals are stable over time.** It takes into account the individual responses of participants, which are then related across time points. Longitudinal reliability at the individual level inherently includes the group level, as it is calculated for the sample as a whole, but the individual responses are central to the calculation. |
| • *Intraclass correlation coefficients (ICCs)* | 'ICC coefficients quantify the extent to which multiple measurements for each individual (within individuals) are statistically similar enough to discriminate between individuals' (*Aldridge et al., 2017*). Here, we calculated two types of ICCs, namely **absolute agreement** and **consistency**. To illustrate the difference between absolute agreement and consistency in a short example (*Koo and Li, 2016*), consider an interrater reliability study with two raters: Consistency indicates the extent to which the score of one rater ($y$) is equal to the score of another rater ($x$) plus a systematic error ($c$) (i.e., $y = x + c$). In contrast, absolute agreement indicates to which degree $y$ equals $x$. As 'two raters' can be replaced by 'two time points' and individual responses were taken into account here, we used ICCs to determine longitudinal reliability at the individual level. |
| • *Within- and between-subject similarity* | Similarity analyses provide information on the extent to which trial-by-trial responses of one individual at one time point are comparable to responses of<br>• the same individual at a later time point (i.e., within-subject similarity) and<br>• all other individuals at a later time point (i.e., between-subject similarity).<br><br>Comparisons of within- and between-subject similarity were used here to determine longitudinal reliability at the individual level. |
| • *Overlap at the individual level (applied for BOLD fMRI only)* | Overlap at the individual level reflects the **degree of overlap of significant voxels** between both time points **for single subject-level** activation patterns. |
| **Longitudinal reliability at the group level** | Longitudinal reliability at the group level indicates to which degree **responses within the group as a whole are stable over time.** More precisely, longitudinal reliability at the group level relies on first averaging all individuals responses for each trial (for SCR) or voxel (for fMRI) yielding a group average for each trial/voxel. These are then related across time points, that is the calculation is carried out using the trial-wise (for SCR) or voxel-wise (for fMRI) group averages. |
| • *Overlap at the group level (applied for BOLD fMRI only)* | Overlap at the group level reflects the **degree of overlap of significant voxels** between both time points **for aggregated group-level** activations. |

*Table 1 continued on next page*

*Table 1 continued*

**(B)**

| | Measure | Internal consistency | Longitudinal reliability at the individual level | | | Longitudinal reliability at the group level | Cross-phases predictability |
|---|---|---|---|---|---|---|---|
| | | | ICCs | Within- and between-subject similarity | Overlap | Overlap (BOLD fMRI) or R squared (SCR) | |
| **Included time points** | All | T0 and T1 separately | T0 and T1 | T0 and T1 | T0 and T1 | T0 and T1 | T0 |
| | SCR | CS+, CS−, CS discrimination, US | CS+, CS−, CS discrimination, US' | CS+, CS−, CS discrimination, US | — | CS+, CS−, CS discrimination, US | CS+, CS−, CS discrimination |
| | Fear ratings | — | CS+, CS−, CS discrimination, US' | — | — | — | CS+, CS−, CS discrimination |
| | BOLD fMRI | — | CS discrimination[†] | CS discrimination[†] | CS discrimination[†] | CS discrimination[†] | CS+, CS−, CS discrimination |
| **Included stimuli** | SCR | Entire phases (ACQ, EXT, RI-Test; except first trials of ACQ and EXT) | CS+, CS−, and CS discrimination: average ACQ, last two trials ACQ[‡], first trial EXT[§], average EXT[¶], last two trials EXT[‡¶], first trial RI-Test[§] US: average RI | Average ACQ[**], average EXT | — | Average ACQ, average EXT | Average ACQ, last two trials ACQ[‡], first trial EXT[§], average EXT, last two trials EXT[‡¶], first trial RI-Test[§] |
| **Phase operationalizations** | Fear ratings | — | CS+, CS−, and CS discrimination: post–pre ACQ, post ACQ, pre EXT, post EXT, first trial RI-Test US: post RI-Test | — | — | — | post–pre ACQ, post ACQ, pre EXT, pre–post EXT, post EXT, first trial RI-Test |
| | BOLD fMRI[††] | — | Average ACQ, average EXT | Average ACQ, average EXT | Average ACQ, average EXT | Average ACQ, average EXT | Average ACQ, average EXT |
| **Transformations[‡‡]** | SCR | None, log-transformation[§§], log-transformation and range correction[¶¶] | None, log-transformation[§§], log-transformation and range correction[¶¶] | None[***] | — | None, log-transformation[§§], log-transformation and range correction[¶¶] | None, log-transformation[§§], log-transformation and range correction[¶¶] |
| | Fear ratings | — | None | — | — | — | None |
| | BOLD fMRI | — | None | None | None | None | None |

*Table 1 continued on next page*

*Table 1 continued*

**(B)**

| Measure | Internal consistency | Longitudinal reliability at the individual level | | | Longitudinal reliability at the group level | | Cross-phases predictability |
|---|---|---|---|---|---|---|---|
| | | ICCs | Within- and between-subject similarity | Overlap | Overlap | Overlap (BOLD fMRI) or R squared (SCR) | |
| **Ordinal ranking†††** | | | | | | | |
| SCR | No ranking | No ranking‡‡‡ | No ranking | – | No ranking | | No ranking and ordinal ranking§§§ |
| Fear ratings | – | No ranking‡‡‡ | – | – | – | | No ranking and ordinal ranking |
| BOLD fMRI | – | No ranking | No ranking | No ranking | No ranking | | No ranking |

The specifications we used here are exemplary and are not intended to cover all possible data specifications. Note that internal consistency, within- and between-subject similarity and reliability at the group level could not be calculated for fear ratings due to the limited number of trials. ACQ = acquisition training, EXT = extinction training, RI = reinstatement training, RI-Test = reinstatement-test.

*Non-pre-registered ICCs for SCRs to the USs and US aversiveness ratings were calculated as we considered these informative.

†For BOLD fMRI, ICCs were calculated only for CS discrimination and not for CS+ and CS− given the fact that the calculations are based on first-level T contrast maps and contrasts against baseline are not optimal.

‡In addition to the averaged acquisition and extinction training performance, the last two SCR trials of acquisition (pre-registered) and extinction training (not pre-registered) were separated from the previous trials and averaged as equivalent to the post-acquisition/-extinction ratings. The first extinction trial was taken into account separately as fear recall.

§Fear recall and reinstatement-test were operationalized as the first extinction training trial and the first reinstatement-test trial (since the reinstatement effect fades away relatively quickly, *Haaker et al., 2014*), respectively.

¶The operationalization of extinction training as the last two trials was not pre-registered and included for completeness. In cases where phase operationalizations included more than one SCR trial, trials were averaged.

**Note that reliability at a group level for SCRs during reinstatement-test was not calculated as correlations between two SCR data points are not meaningful.

††fMRI data for the reinstatement-test were not analyzed in the current study since data from a single trial do not provide sufficient power.

‡‡The pre-registered transformation types were identified to be typically employed data transformations in the literature by for example *Sjouwerman et al., 2022* who also pre-registered these transformation types.

§§Raw SCR amplitudes were log-transformed by taking the natural logarithm to normalize the distribution (*Levine and Dunlap, 1982*).

¶¶Log-transformed SCR amplitudes were range corrected by dividing each individual SCR trial by the maximum SCR trial across all CS and US trials. Due to potentially different response ranges, the maximum SCR trial was determined separately for experimental days as recommended by *Lonsdorf et al., 2017a*. Range correction was recommended to control for interindividual variability (*Lykken, 1972; Lykken and Venables, 1971*).

***We also carried out similarity analyses for log-transformed as well as for log-transformed and range corrected data. However, results were almost identical to the results from the raw data. For reasons of space, we only report results for raw data.

†††Ranking of the data was included to investigate to which degree individuals occupy the same ranks at both time points as pre-registered or put differently, whether the quality of predictions changes when the predictions were not based on the absolute values but on a coarser scale.

‡‡‡As opposed to what was pre-registered, in ICC analyses, we included non-ranked data only as closer inspection of the conceptualization of $ICC_{con}$ revealed that it would be redundant to calculate both $ICC_{abs}$ and $ICC_{con}$ with ranked and non-ranked data as $ICC_{con}$ itself ranks the data.

§§§Ranks of SCRs were built upon raw, log-transformed as well as log-transformed and range corrected values.

than for extinction training (SCRs: *Fredrikson et al., 1993*; *Zeidan et al., 2012*), but comparable to generalization (*Cooper et al., 2022b*; *Torrents-Rodas et al., 2014*). Moreover, it appeared higher for extinction training than for reinstatement-test (for BOLD fMRI but not ratings: *Ridderbusch et al., 2021*) and higher for CS+ than CS− responses (SCRs: *Fredrikson et al., 1993*) and CS discrimination (ratings and BOLD fMRI: *Ridderbusch et al., 2021*; SCRs: *Zeidan et al., 2012*).

However, it is difficult to extract a comprehensive picture from these five studies as they differ substantially in sample size (*N* = 18–100), paradigm specifications, experimental phases reported, outcome measures, time intervals, and employed reliability measures (see *Supplementary file 1*).

Given that the predominance of research on group-level generic mechanisms in fear conditioning research, it is even more surprising that, to our knowledge, no study to date has investigated longitudinal reliability at the group level and only few studies have (*Fredrikson et al., 1993*) targeted internal consistency (i.e., the degree to which all test items capture the same construct, see *Table 1*). More precisely, longitudinal reliability at the group level indicates the extent to which responses averaged across the group as a whole are stable over time, which is important to establish when investigating basic, generic principles such as the impact of experimental manipulations. Even though it has to be acknowledged that the group average is not necessarily representative of any individual in the group and the same group average may arise from different and even opposite individual responses at both time points in the same group, group-level reliability is important to establish in addition to individual-level reliability. Group-level reliability is relevant not only to work focusing on the understanding of general, generic processes but also for questions about differences between two groups of individuals such as patients vs. controls (e.g., see meta-analyses of *Cooper et al., 2022a*; *Duits et al., 2015*). Of note, many fear conditioning paradigms were initially developed to study general group-level processes and to elicit robust group effects (*Lonsdorf and Merz, 2017b*). Hence, it is important to investigate both group- and individual-level reliability given the challenges of attempts to employ cognitive tasks that were originally designed to produce robust group effects in individual difference research (*Elliott et al., 2020*; *Hedge et al., 2018*; *Parsons, 2020*; *Parsons et al., 2019*).

As pointed out above, individual-level reliability is a prerequisite for individual-level predictions such as treatment outcomes. Since the different experimental phases of fear conditioning paradigms serve as experimental models for the development, treatment, and relapse of anxiety- and stress-related disorders, it is also an important question whether responding across phases can be reliably predicted at the individual level. Interestingly, it is often implicitly assumed that responding in one experimental phase reliably predicts responding in a subsequent phase (e.g., see *Milad et al., 2009*; critically discussed in *Lonsdorf et al., 2019a*) even though empirical evidence is lacking. As a result it has been suggested to routinely 'correct for responding' during fear acquisition training when studying performance in later experimental phases such as extinction training or retention/RoF test (critically discussed in *Lonsdorf et al., 2019a*). However, empirical evidence on this cross-phases predictability (for definition and terminology, see *Table 1*) is scarce to date.

Evidence from experimental work on cross-phase predictability in rodents and humans is mixed. In rodents, freezing during acquisition training and 24-hrs-delayed extinction training were uncorrelated (*Plendl and Wotjak, 2010*) and responding during extinction training did not predict extinction retention (i.e., lever-pressing suppression: *Bouton et al., 2006*; or freezing behavior: *Shumake et al., 2014*). Similarly, in humans, extinction performance (FPS, SCRs, and US expectancy ratings) did not predict performance at 24-hrs-retention test (*Prenoveau et al., 2013*). Yet, a computational modeling approach suggests that the mechanism of extinction learning (i.e., the formation of a new extinction memory trace in comparison to an update of the original fear memory trace) predicts the extent of spontaneous recovery in SCRs (*Gershman and Hartley, 2015*).

Also evidence from work in patient samples is mixed (for a review, see *Craske et al., 2008*). The extent of fear reduction within therapeutic sessions was unrelated to overall treatment outcome in some studies (*Kozak et al., 1988*; *Pitman et al., 1996*; *Riley et al., 1995*), while others observed an association (*Foa et al., 1983*). Similarly, significant correlations of fear reduction between therapeutic sessions with treatment outcome were observed for reported distress (*Rauch et al., 2004*) and heart rate, but not for SCR (*Kozak et al., 1988*; *Lang et al., 1970*) and for self-reported fear post treatment, but not at follow-up (*Foa et al., 1983*). In addition, evidence that responding in different phases is related comes from pharmacological manipulations with the cognitive enhancer D-cycloserine which facilitates learning and/or consolidation. D-cycloserine promoted long-term extinction retention

(*Rothbaum et al., 2014*; *Smits et al., 2013a*; *Smits et al., 2013b*) only if within-session learning was achieved.

With this pre-registered study, we follow the call for a stronger appreciation and more systematic investigations of measurement reliability (*Zuo et al., 2019*). We address longitudinal reliability and internal consistency as well as predictability of cross-phase responding in SCRs, fear ratings, and the BOLD response. For this purpose, we reanalyzed data from 120 participants that underwent a differential fear conditioning paradigm twice (at time points T0 and T1, 6 months apart) – with habituation and acquisition training on day 1 and extinction, reinstatement and reinstatement-test on day 2 to allow for fear memory consolidation prior to extinction. Part of the data have been used previously in method focused work (*Kuhn et al., 2022*; *Lonsdorf et al., 2022*; *Lonsdorf et al., 2019a*; *Sjouwerman et al., 2022*) and work investigating the association of conditioned responding with brain morphological measures (*Ehlers et al., 2020*).

Specifically, we (1) estimated internal consistency of SCRs at both time points and (2) systematically assessed longitudinal reliability of SCRs, fear ratings and BOLD fMRI at the individual level by calculating ICCs. This was complemented by investigations of response similarity (SCR and BOLD fMRI) and the degree of overlap of activated voxels at both time points (BOLD fMRI) as additional measurements of longitudinal reliability at the individual level that allow for a more detailed picture than the coarser ICCs (see *Table 1* for terminology and definitions). We also (3) assessed whether SCR and BOLD fMRI show longitudinal reliability at the group level. Finally, we (4) investigated if individual level responding during an experimental phase is predictive of individual-level responding during subsequent experimental phases. All hypotheses are tested across different pre-registered data specifications to account for procedural heterogeneity in the literature (see *Supplementary file 1*): More precisely, we follow a pre-registered multiverse-inspired approach and include (1) responses to the CS+, CS−, US, and CS discrimination, (2) different phase operationalizations, (3) different data transformations none, log-transformed, log-transformed and range-corrected, and (4) ordinally ranked vs. non-ranked data (for justification of these choices, see *Table 1*). We acknowledge that the specifications used here are not intended to cover all potentially meaningful combinations as in a full multiverse study (*Lonsdorf et al., 2022*; *Sjouwerman et al., 2022*; *Steegen et al., 2016*) but can be viewed as a manyverse (*Kuhn et al., 2022*) in which we a priori pre-registered a number of meaningful combinations.

## Results

For a comprehensive overview of the different reliability measures used here and of the analyses conducted, see *Table 1*.

### Satisfactory internal consistency

To assess internal consistency of SCRs, trials were split into odd and even trials (i.e., odd–even approach), averaged for each individual subject and then correlated (Pearson's correlation coefficient). This was done separately for each time point and experimental phase. Internal consistency at T0 (see *Figure 1A*) and T1 (see *Figure 1B*) of raw SCRs to the CS+ and CS− ranged from 0.54 to 0.85 and for raw SCRs to the US from 0.91 to 0.94 for all phases. In comparison, internal consistency was lower for CS discrimination with values ranging from −0.01 to 0.60. Log-transformation did not impact internal consistency but log-transformation in combination with range correction largely resulted in reduced reliability (see *Figure 1—figure supplement 1*).

### Longitudinal reliability at the individual level

Longitudinal reliability at the individual level refers to the time stability of individual responses which we assessed through several measures (see *Table 1*).

As a first measure, absolute agreement ICCs (ICC$_{abs}$) and consistency ICCs (ICC$_{con}$) were calculated across both time points (T0, T1) for all data specifications (see *Figure 1*) while for BOLD fMRI these were only calculated for CS discrimination (see Materials and methods for justification). While ICC$_{abs}$ refers to the extent to which measurements at T0 correspond with measurements at T1 in absolute terms, ICC$_{con}$ allows for deviations at T1 due to systematic error (*Koo and Li, 2016*).

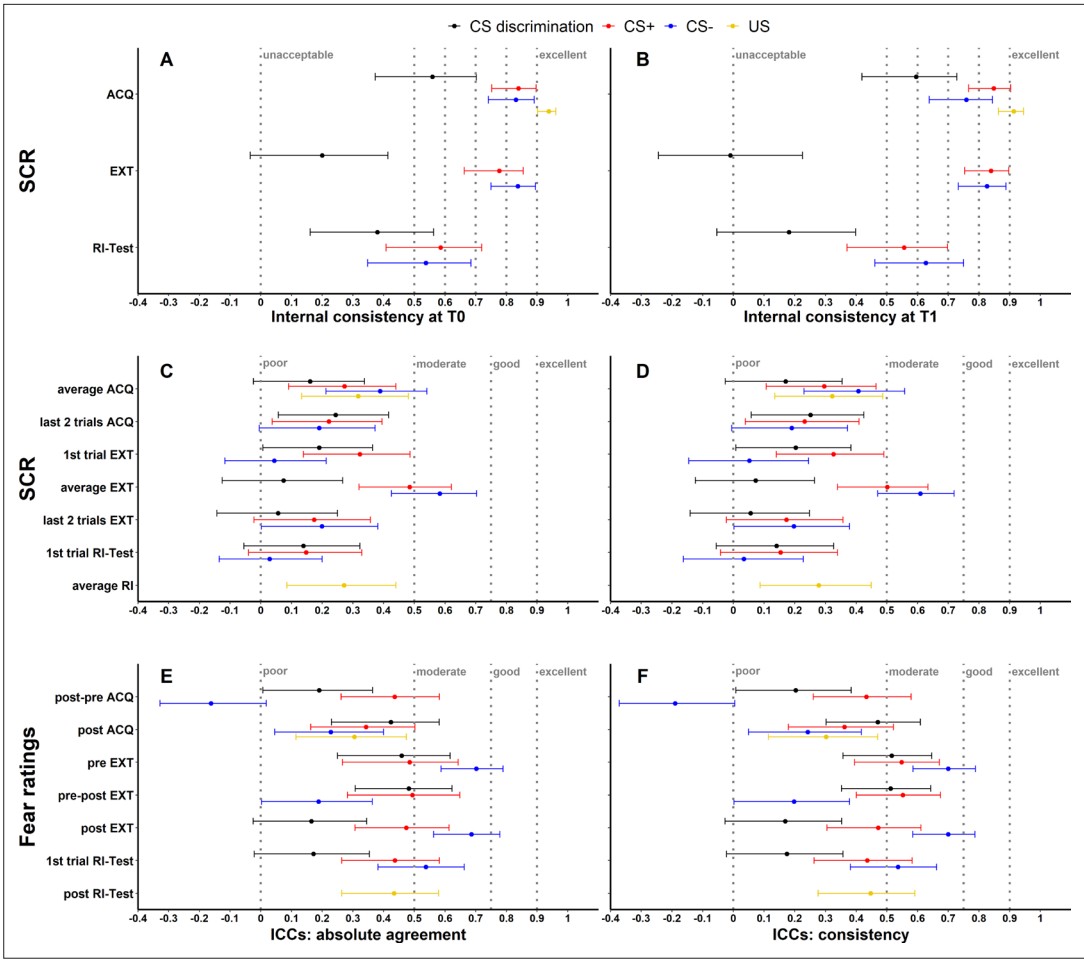

**Figure 1.** Illustration of internal consistency for skin conductance responses (SCRs) at T0 (**A**) and T1 (**B**) as well as ICC$_{abs}$ and ICC$_{con}$ for SCRs (**C, D**) and fear ratings (**E, F**) color coded for stimulus type. Internal consistency indicates the reliability of responses within each time point, while intraclass correlation coefficients (ICCs) indicate the reliability across both time points. Note that assessment of internal consistency was not possible for fear ratings as only two ratings (pre, post) were available. Error bars represent 95% confidence intervals and indicate significance, when zero is not included in the interval. The y-axis comprises the different phases or phase operationalizations. In the literature, internal consistency is often interpreted using benchmarks (**Kline, 2013**) for unacceptable (<0.5), poor (>0.5 but <0.6), questionable (>0.6 but <0.7), acceptable (>0.7 but <0.8), good (>0.8 but <0.9), and excellent (≥0.9). Common benchmarks in the literature for ICCs are poor (<0.5), moderate (>0.5 but <0.75), good (>0.75 but <0.9), and excellent (≥0.9) (**Koo and Li, 2016**). These benchmarks are included here to provide a frame of reference but we point out that these benchmarks are arbitrary and most importantly derived from psychometric work on trait self-report measures and should hence not be overinterpreted in the context of responding in experimental paradigms which bear more sources of noise (**Parsons, 2020**). ACQ = acquisition training, EXT = extinction training, RI = reinstatement, RI-Test = reinstatement-test, pre = prior to the experimental phase, post = subsequent to the experimental phase.

The online version of this article includes the following figure supplement(s) for figure 1:

**Figure supplement 1.** Illustration of (**A, B**) internal consistency for log-transformed (log) as well as (**C, D**) log-transformed and range corrected (log rc) skin conductance responses (SCRs) at T0 and T1 color coded for stimulus type.

**Figure supplement 2.** Illustration of (**A, B**) intraclass correlation coefficients (ICCs) of log-transformed (log) as well as (**C, D**) log-transformed and range corrected (log, rc) skin conductance responses (SCRs) color coded for stimulus type.

**Figure supplement 3.** Illustration of ICC$_{abs}$ of trial-by-trial raw skin conductance responses (SCRs) for phases (A–D: Acquisition, E–G: Extinction, H–J: Reinstatement-Test, K: Reinstatement) and stimulus types separately.

*Figure 1 continued on next page*

*Figure 1 continued*

**Figure supplement 4.** Illustration of ICC<sub>con</sub> of trial-by-trial raw skin conductance responses (SCRs) for phases (A–D: Acquisition, E–G: Extinction, H–J: Reinstatement-Test, K: Reinstatement) and stimulus types separately.

**Figure supplement 5.** Illustration of ICC<sub>abs</sub> of trial-by-trial log-transformed skin conductance responses (SCRs) for phases (A–D: Acquisition, E–G: Extinction, H–J: Reinstatement-Test, K: Reinstatement) and stimulus types separately.

**Figure supplement 6.** Illustration of ICC<sub>con</sub> of trial-by-trial log-transformed skin conductance responses (SCRs) for phases (A–D: Acquisition, E–G: Extinction, H–J: Reinstatement-Test, K: Reinstatement) and stimulus types separately.

**Figure supplement 7.** Illustration of ICC<sub>abs</sub> of trial-by-trial log-transformed and range corrected skin conductance responses (SCRs) for phases (A–D: Acquisition, E–G: Extinction, H–J: Reinstatement-Test, K: Reinstatement) and stimulus types separately.

**Figure supplement 8.** Illustration of ICC<sub>con</sub> of trial-by-trial log-transformed and range corrected skin conductance responses (SCRs) for phases (A–D: Acquisition, E–G: Extinction, H–J: Reinstatement-Test, K: Reinstatement) and stimulus types separately.

Note that internal consistency and ICCs for SCRs are shown for raw data only. Results of log-transformed as well as log-transformed and range corrected data are presented in *Figure 1—figure supplement 1* and *Figure 1—figure supplement 2* for completeness.

## SCR and fear ratings

Across data specifications, $ICC_{abs}$ and $ICC_{con}$ ranged from 0.03 to 0.58 and 0.03 to 0.61 for SCRs and from -0.16 to 0.70 as well as from -0.19 to 0.70 for fear ratings respectively (see *Figure 1*, for detailed results see also *Supplementary file 3* and *Supplementary file 4*). ICCs for log-transformed and raw SCRs were similar (see *Figure 1—figure supplement 2A-B*) while log-transformation and range correction resulted in increased reliability for some data specifications (e.g., CS+ and CS- responses averaged across acquisition training, see *Figure 1—figure supplement 2C-D*) but in reduced reliability for others (e.g., CS- responses during fear recall, i.e., the first extinction trial).

Exploratory, non-pre-registered analyses of trial-by-trial SCRs revealed, overall, only minor changes in ICCs upon stepwise inclusion of additional SCR trials (see *Figure 1—figure supplements 3–8*) with few exceptions: Including more trials resulted in an increase of ICC point estimates for SCRs to the CS+ and CS− during acquisition (log-transformed and range corrected data) and extinction training (all transformation types). Note, however, that this was – at large – only statistically significant when comparing ICCs based on the first (i.e., single trial at T0 and T1) and the maximum number of trials (as indicated by non-overlapping 95% confidence interval [CI] error bars). Interestingly, ICC point estimates for reinstatement-test (all transformation types) were numerically lower with an increasing number of trials, likely because of the transitory nature of the reinstatement effect (*Haaker et al., 2014*).

## BOLD fMRI

For BOLD fMRI, both ICC types suggest rather limited reliability for CS discrimination during acquisition (both $ICC_{abs}$ and $ICC_{con}$ = 0.17) and extinction training (both $ICC_{abs}$ and $ICC_{con}$ = 0.01). For individual regions of interest (ROIs: anterior insula, amygdala, hippocampus, caudate nucleus, putamen, pallidum, nucleus accumbens [NAcc], thalamus, dorsal anterior cingulate cortex [dACC], dorsolateral prefrontal cortex [dlPFC], and ventromedial prefrontal cortex [vmPFC]), ICCs were even lower (all ICCs ≤0.001; for full results see *Supplementary file 5*).

## Higher within- than between-subject similarity in BOLD fMRI but not SCRs

While ICCs provide information on the absolute quantity of longitudinal reliability at the individual level, comparison of within- and between-subject similarity as a complementary measure of longitudinal reliability at the individual level (see *Table 1*) reflects the extent to which responses in SCR and BOLD activation of one individual at T0 were more similar to themselves at T1 than to other individuals at T1 (see *Figures 2 and 3*).

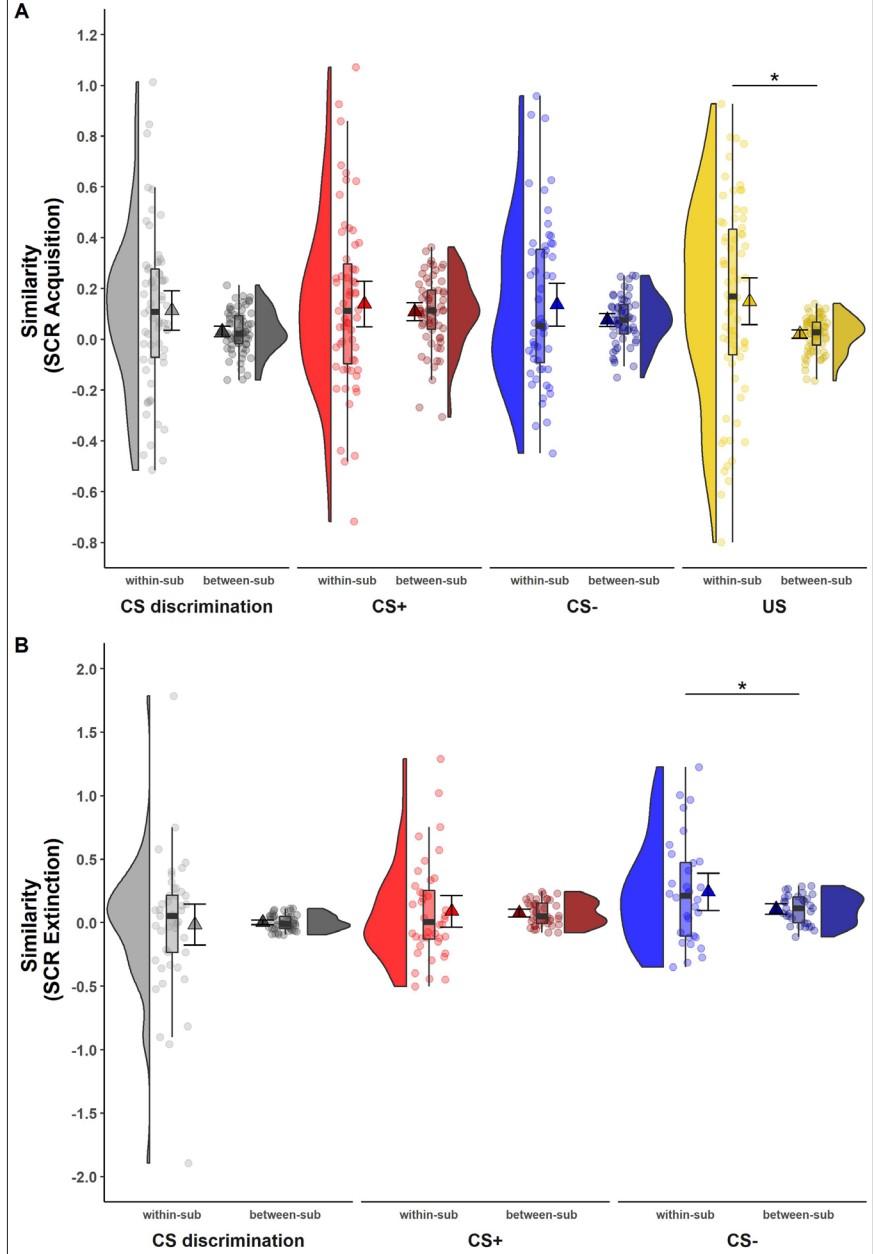

**Figure 2.** Illustration of within- and between-subject similarity for raw skin conductance responses (SCRs) during (**A**) acquisition and (**B**) extinction training separately for CS discrimination (gray), CS+ (red), CS− (blue), and unconditioned stimulus (US) responses (yellow). Results for log-transformed as well as log-transformed and range corrected SCRs were almost identical to the results from raw data and are hence not reported here. Single data points represent Fisher *r*-to-*z* transformed correlations between single trial SCRs of each subject at T0 and T1 (within-subject similarity) or averaged *r*-to-*z* transformed correlations between single trial SCRs of one subject at T0 and all other subjects at T1 (between-subject similarity). Triangles represent mean correlations, corresponding error bars represent 95% confidence intervals. Boxes of boxplots represent the interquartile range (IQR) crossed by the median as bold line, ends of whiskers represent the minimum/maximum value in the data within the range of 25th/75th percentiles ±1.5 IQR. Distributions of the data are illustrated by densities next to the boxplots. One data point had a similarity above 3.5 (within-subject similarity of SCRs to the CS+) and is not shown in the figure. *p < 0.05. Note that the variances differ strongly between within- and between-subject similarity because between-subject similarity is based on correlations averaged across subjects, whereas within-subject similarity is based on non-averaged correlations calculated for each subject. Note also that similarity calculations were based on different sample sizes for acquisition and extinction training and CS discrimination as well as SCRs to the CS+, CS−, and US, respectively (for details, see Materials and methods). within-sub = within-subject; between-sub = between-subject.

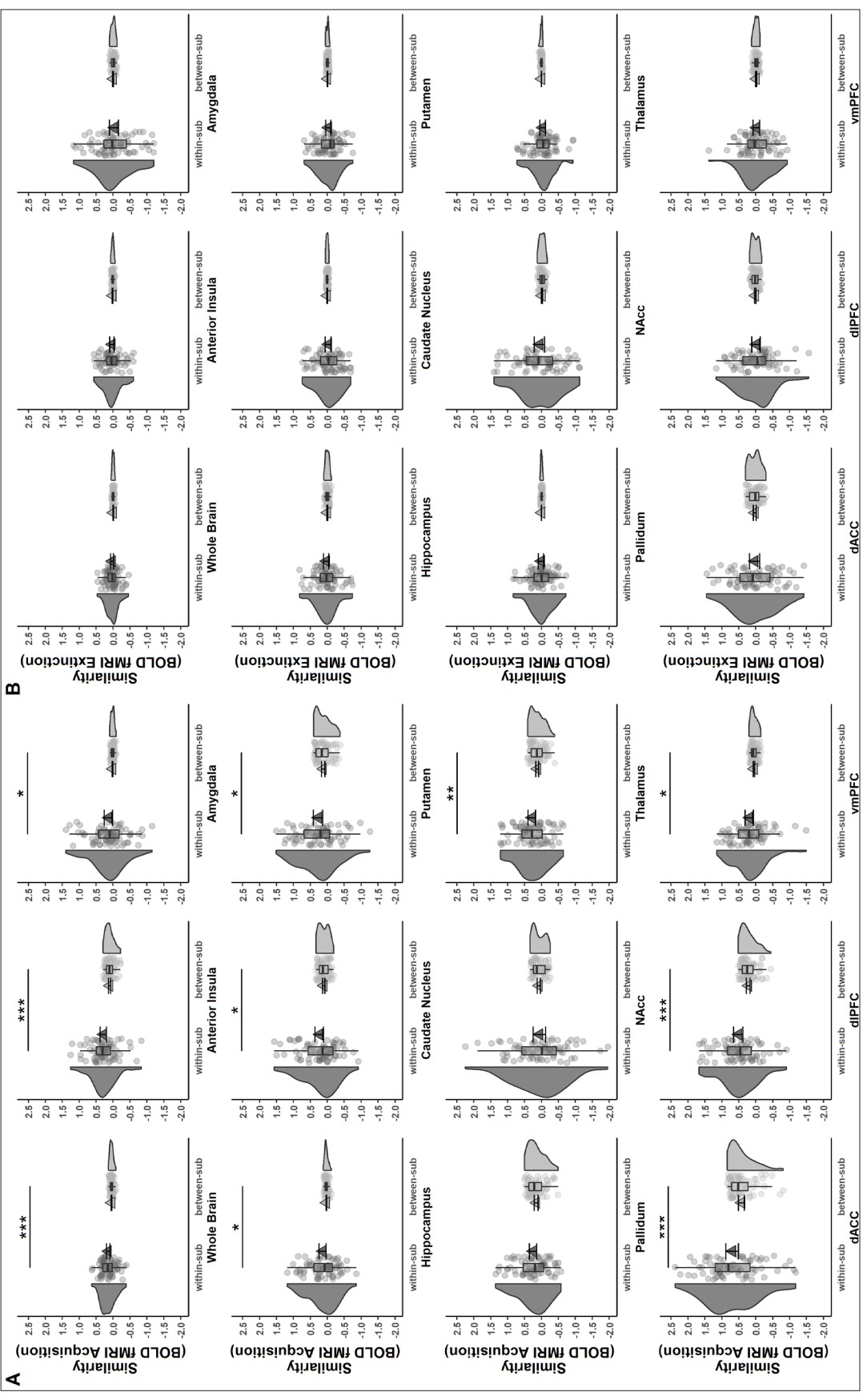

**Figure 3.** Acquisition (**A**) and extinction (**B**) training within- and between-subject similarities (Fisher *r-to-z* transformed) of voxel-wise brain activation patterns (based on beta maps) for CS discrimination at T0 and T1 for the whole brain and different regions of interest (ROIs). Triangles represent mean correlations, corresponding error bars represent 95% confidence intervals. Single data points represent Fisher *r-to-z* transformed correlations between the first-level response patterns of brain activation of each subject at T0 and T1 (within-subject similarity) or averaged *r-to-z* transformed correlations between the first-level response patterns of brain activation of one subject at T0 and all other subjects at T1 (between-subject similarity). Boxes of boxplots represent the interquartile range (IQR) crossed by the median as bold line, ends of whiskers represent the minimum/maximum value in the data within the range of 25th/75th percentiles ±1.5 IQR. Distributions of the data are illustrated with densities next to the boxplots. fMRI data for the reinstatement-test were not analyzed in the current study since data from a single trial do not provide sufficient power. *p < 0.05, **p < 0.01, ***p < 0.001. NAcc = nucleus accumbens; dACC = dorsal anterior cingulate cortex; dlPFC = dorsolateral prefrontal cortex; vmPFC = ventromedial prefrontal cortex; within-sub = within-subject; between-sub = between-subject.

## SCR

For SCRs, within-subject similarity (i.e., within-subject correlation of trial-by-trial SCR across time points) and between-subject similarity (i.e., correlation of trial-by-trial SCR between one individual at T0 and all other individuals at T1; see *Figure 2*) did not differ significantly for most data specifications. This was true for CS discrimination (*t*(64) = 1.78, p = 0.079, *d* = 0.22) as well as for SCRs to the CS+ (*t*(61) = 0.84, p = 0.407, *d* = 0.11) and CS− (*t*(55) = 1.50, p = 0.138, *d* = 0.20) during acquisition training and for CS discrimination (*t*(44) = −0.23, p = 0.823, *d* = −0.03) and SCRs to the CS+ (*t*(39) = 0.25, p = 0.801, *d* = 0.04) during extinction training. This indicates that SCRs of one particular individual at T0 were mostly not more similar to their own SCRs than to those of other individuals at T1. The only exceptions where within-subject similarities were significantly higher than between-subject similarity were SCRs to the US during acquisition training (*t*(70) = 2.54, p = 0.013, *d* = 0.30) and to the CS− during extinction training (*t*(31) = 2.05, p = 0.049, *d* = 0.36). Note, however, that within-subject similarity had a very wide spread pointing to substantial individual differences (while this variance is removed in calculations of between-subject similarity).

## fMRI data

In contrast to what was observed for SCRs, within-subject similarity was significantly higher than between-subject similarity in the whole brain (p < 0.001) and most of the ROIs for fear acquisition training (see *Figure 3A* and *Supplementary file 6*). This suggests that while absolute values for similarity might be low, individual brain activation patterns during fear acquisition training at T0 were – at large – still more similar to the same subject's activation pattern at T1 than to any others at T1. For extinction training, however, no significant differences between within- and between-subject similarity were found for any ROI or the whole brain (all p's > 0.306; see *Figure 3B* and *Supplementary file 6*).

**Table 2.** Overlap in significantly activated voxels at the individual and group level across both time points for CS discrimination.

| Level | Phase | Coeff. | Whole brain | Insula | Amygdala | Hippocampus | Caudate | Putamen | Pallidum | Accumbens | Thalamus | dACC | dlPFC | vmPFC |
|---|---|---|---|---|---|---|---|---|---|---|---|---|---|---|
| | | | | | | | | | | ROI | | | | |
| | | Jaccard | 0.076 | 0.075 | 0.011 | 0.012 | 0.039 | 0.037 | 0.018 | 0.017 | 0.033 | 0.132 | 0.080 | 0.039 |
| | Acq | Dice | 0.131 | 0.121 | 0.018 | 0.021 | 0.057 | 0.058 | 0.029 | 0.024 | 0.055 | 0.189 | 0.118 | 0.061 |
| | | Jaccard | 0.007 | 0.001 | 0.000 | 0.000 | 0.001 | 0.000 | 0.000 | 0.000 | 0.001 | 0.003 | 0.001 | 0.005 |
| (A) Individual | Ext | Dice | 0.014 | 0.001 | 0.000 | 0.000 | 0.001 | 0.000 | 0.000 | 0.001 | 0.001 | 0.006 | 0.002 | 0.009 |
| | | Jaccard | 0.620 | 0.595 | 0.294 | 0.323 | 0.613 | 0.740 | 0.747 | 0.441 | 0.834 | 0.898 | 0.895 | 0.045 |
| | Acq | Dice | 0.765 | 0.745 | 0.448 | 0.472 | 0.760 | 0.847 | 0.855 | 0.595 | 0.910 | 0.946 | 0.944 | 0.086 |
| | | Jaccard | 0.057 | 0.000 | 0.000 | 0.000 | 0.000 | 0.000 | 0.000 | 0.000 | 0.000 | 0.044 | 0.014 | 0.000 |
| (B) Group | Ext | Dice | 0.108 | 0.000 | 0.000 | 0.000 | 0.000 | 0.000 | 0.000 | 0.000 | 0.000 | 0.085 | 0.028 | 0.000 |

Note. Results are shown for the whole brain as well as for selected regions of interest (ROIs) for fear acquisition training and extinction training. Both coefficients range from 0 (no overlap) to 1 (perfect overlap). Note that the Jaccard can be interpreted as % (**Maitra, 2010**). NAcc = nucleus accumbens; dACC = dorsal anterior cingulate cortex; dlPFC = dorsolateral prefrontal cortex; vmPFC = ventromedial prefrontal cortex.

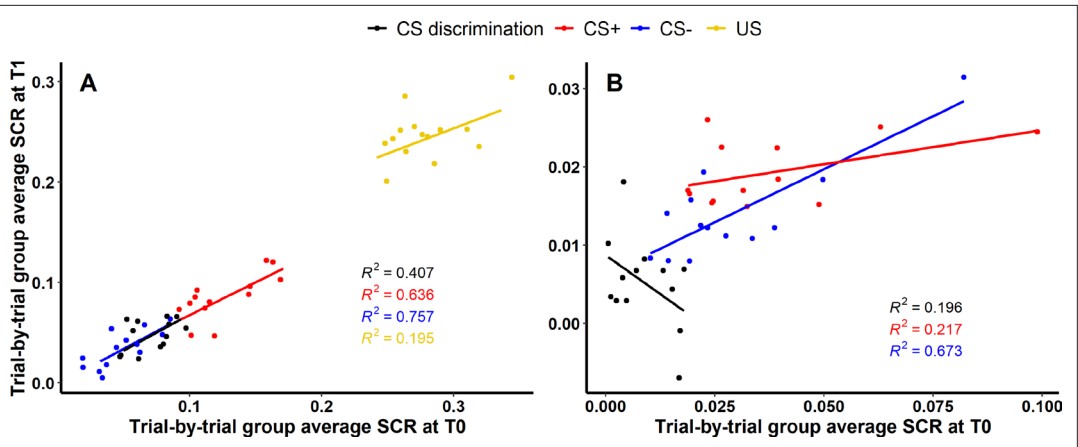

**Figure 4.** Scatter plots illustrating longitudinal reliability at the group level during (**A**) acquisition and (**B**) extinction training for raw skin conductance responses (SCRs) (in μS). Results for log-transformed as well as log-transformed and range corrected data are presented in *Figure 4—figure supplement 1*. Longitudinal reliability at the group level refers to the extent of explained variance in linear regressions comprising SCRs at T0 as independent and SCRs at T1 as dependent variable. Results are shown for trial-by-trial group average SCRs to the CS+ (red), CS− (blue), the unconditioned stimulus (US; yellow), and CS discrimination (black). Single data points represent pairs of single trials at T0 and T1 averaged across participants. Note that no US was presented during extinction training and hence, no reliability of the US is shown in (**B**).

The online version of this article includes the following figure supplement(s) for figure 4:

**Figure supplement 1.** Scatter plots illustrating longitudinal reliability at the group level during (**A, C**) acquisition and (**B, D**) extinction training for log-transformed (**A, B**) as well as log-transformed and range corrected (**C, D**) skin conductance responses (SCRs).

## Low overlap at the individual level between both time points

As opposed to similarity measures (see above) which reflect the correlation of activated voxels between time points, overlap at the individual level denotes the degree of overlap of significantly activated voxels.

The overlap at the individual level was low with the Jaccard coefficient indicating 7.60% and 0.70% whole brain overlap for acquisition and extinction training, respectively (see *Table 2A*). Of note, individual values ranged from 0% to 39.65% overlap during acquisition, suggesting large interindividual differences in overlap.

While overlap during acquisition for individual ROIs was comparable to the whole brain, Jaccard and Dice coefficients indicate close to 0 overlap at extinction (see *Table 2A*).

## Robust longitudinal reliability at the group level

While longitudinal reliability at the individual level relies on (mean) individual subject responding at both time points, longitudinal reliability at the group level relies on the percentage of explained variance of group averaged trials at T1 by group averaged trials at T0 (i.e., *R* squared for SCR) or the degree of group level overlap of significant voxels expressed as Dice and Jaccard indices (i.e., BOLD fMRI).

### SCR

For acquisition training (see *Figure 4A*), 40.66% ($F(1, 11) = 7.54$, p = 0.019), 63.59% ($F(1, 11) = 19.21$, p = 0.001) and 75.67% ($F(1, 11) = 34.20$, p < 0.001) of the variance of SCRs at T1 could be explained by SCRs at T0 for CS discrimination, CS+ and CS−, respectively, indicating robust longitudinal reliability of SCRs at the group level for CS responding during acquisition. Interestingly, only 19.53% ($F(1, 12) = 2.91$, p = 0.114) of the variance of SCRs to the US could be explained. For extinction training, in contrast, only 19.58% ($F(1, 11) = 2.68$, p = 0.130) and 21.70% ($F(1, 11) = 3.05$, p = 0.109) of the SCR variance at T1 could be explained by SCRs at T0 for CS discrimination and CS+, respectively, indicating only limited longitudinal reliability at the group level. However, with 67.35% ($F(1, 11) = 22.69$,

p = 0.001) explained variance at T1, longitudinal reliability of SCRs to the CS− appeared to be more robust as compared to CS discrimination and responses to the CS+ (see *Figure 4B*).

## BOLD fMRI

In stark contrast to the low overlap of individual-level activation (see *Table 2A*), the overlap at the group level was rather high with 62.00% for the whole brain and up to 89.80% for individual ROIs (i.e., dACC and dlPFC; Jaccard) for CS discrimination during acquisition training (see *Table 2B*). Similar to what was observed for overlap at the individual level, a much lower overlap for extinction training as compared to acquisition training was observed for the whole brain (5.70% overlap) and all ROIs (all close to zero).

## Cross-phases predictability of conditioned responding

Finally, we investigated if responding in any given experimental phase predicted responding in subsequent experimental phases. To this end, simple linear regressions with robust standard errors were computed for both SCRs and fear ratings and all data specifications (see *Figure 5* and *Supplementary file 7*, *Supplementary file 8*). To approximate these analyses, correlations of patterns of BOLD brain activation between experimental phases were calculated (see *Figure 6*).

### SCR

Stronger CS discrimination in SCRs during (delayed) fear recall (i.e., first trial of extinction training) was significantly predicted by both average and end-point performance (i.e., last two trials) during acquisition training for most data specifications (*Figure 5A*, columns 1 and 2). In contrast, average CS discrimination during extinction training was significantly predicted by acquisition training performance only if data were ordinally ranked (columns 3 and 4). Strikingly, all predictions of extinction end-point performance (columns 5 and 6) as well as performance at reinstatement-test (columns 7–11) were non-significant – irrespective of phase operationalizations and data transformation.

The majority of predictions of SCRs to the CS+ and CS− were significant with few exceptions (see white cells in *Figure 5A*) – irrespective of experimental phases, their operationalization and data transformation. Most non-significant regressions included log-transformed and range corrected data. Strikingly, extinction end-point performance never predicted performance at reinstatement-test – irrespective of data transformation (column 11).

### Fear ratings

Higher ratings for the CS+ as well as higher CS discrimination during acquisition training predicted higher CS+ ratings and CS discrimination at fear recall (*Figure 5B*, columns 1 and 2), extinction training (columns 3 and 4), and at reinstatement-test (columns 7 and 8). Higher responding to the CS+ and higher CS discrimination at fear recall predicted higher responding at reinstatement-test (column 9) – irrespective of data transformations. In contrast, predictions of CS discrimination and CS+ ratings after extinction training were mostly non-significant (columns 5 and 6). Higher CS+ ratings during extinction training significantly predicted higher ratings at reinstatement-test which was not true for CS discrimination (columns 10 and 11).

Higher CS− ratings after acquisition training predicted higher CS− ratings at fear recall as well as after extinction training and CS− ratings after extinction training predicted the performance at reinstatement-test – irrespective of ranking of the data (columns 2, 6, and 11). Furthermore, when based on ordinally ranked data, the difference between ratings prior to and after acquisition predicted CS− ratings at fear recall and CS− ratings after acquisition training predicted the difference between CS− ratings prior to and after extinction training (columns 1 and 4). All other predictions were non-significant.

In sum, all significant predictions observed were positive with weak to moderate associations and indicate that higher responding in preceding phases predicted higher responding in subsequent phases for both SCRs and fear ratings.

### BOLD fMRI

In short, all but one association (CS discrimination in the NAcc) was positive, showing that higher BOLD response during acquisition was associated with higher BOLD responding during extinction

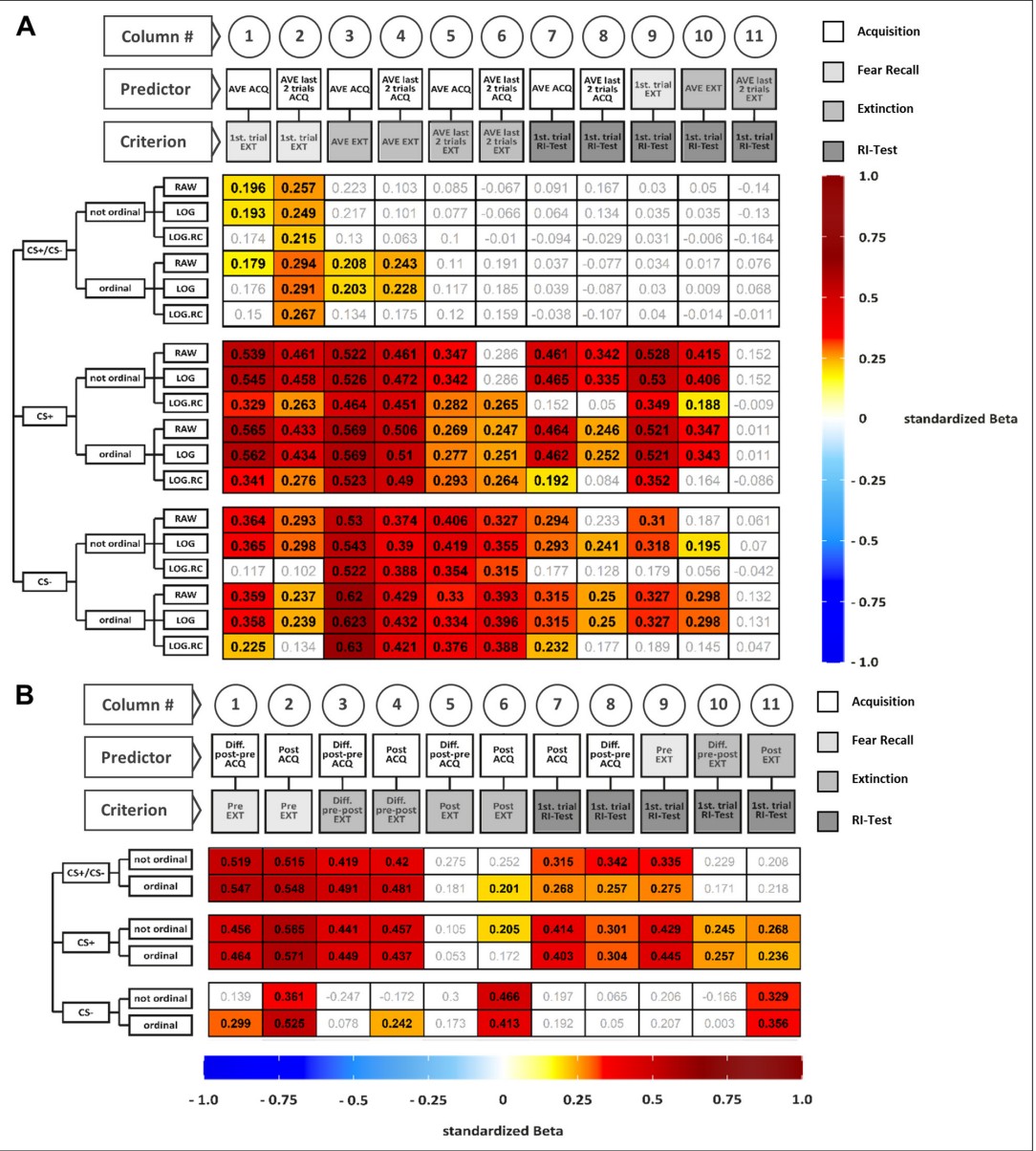

**Figure 5.** Illustration of standardized betas derived from regressions including skin conductance responses (SCRs) (**A**) and fear ratings (**B**) for all data specifications. Colored cells indicate statistical significance of standardized betas, non-colored cells indicate non-significance. Standardized betas are color coded for their direction and magnitude showing positive values from yellow to red and negative values from light blue to dark blue. Darker colors indicate higher betas. On the y-axis, the following data specifications are plotted from left to right: CS type, ranking of the data and transformation of the data. On the x-axis, the following information is plotted: Number of the columns for better orientation, predictor, and criterion included in the regression. For example, the beta value at the top left in (**A**) (i.e., 0.196) is the standardized beta as retrieved from the linear regression including CS discrimination in non-ranked and raw SCRs during average acquisition as predictor and the first extinction trial as criterion. For exploratory non-preregistered regressions including a small manyverse of approximations of SCR extinction training learning rates, see **Figure 5—figure supplement 1**. Tables containing regression parameters beyond the standardized betas depicted in panels A and B are presented in **Supplementary file 7** and **Supplementary file 8**. AVE = average, LOG = log-transformed data, LOG.RC = log-transformed and range corrected data, not ordinal = not ordinally ranked data, ordinal = ordinally ranked data.

The online version of this article includes the following figure supplement(s) for figure 5:

**Figure supplement 1.** As per reviewer's request, we illustrate standardized betas derived from non-pre-registered regressions including skin conductance response (SCR) extinction training learning rates (LR EXT).

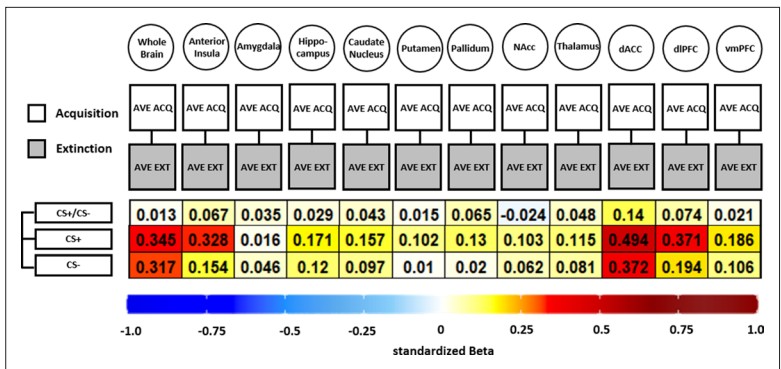

**Figure 6.** Illustration of standardized betas derived from correlation analyses between brain activation patterns during acquisition and extinction training in different regions of interest (ROIs) and different data specifications. Standardized betas are color coded for their direction and magnitude showing positive values from yellow to red and negative values from light blue to dark blue. Darker colors indicate higher betas. NAcc = nucleus accumbens; dACC = dorsal anterior cingulate cortex; dlPFC = dorsolateral prefrontal cortex; vmPFC = ventromedial prefrontal cortex.

training (see *Figure 6*). However, the standardized beta coefficients are mostly below or around 0.3 except for CS+ associations in the dACC, indicating non-substantial associations for all ROIs and CS specifications that were near absent for CS discrimination. Analysis of CS+ and CS− data was included here as the analysis is based on beta maps and not T-maps (as in previous analyses) where a contrast against baseline is not optimal.

## Cross-phases predictability depends on data specifications

Pooled across all other data specifications, some interesting patterns can be extracted: First, standardized betas were significantly lower for raw ($t(65)$ = 8.08, p < 0.001, $d$ = 0.99) and log-transformed ($t(65)$ = 8.26, p < 0.001, $d$ = 1.02) as compared to log-transformed and range corrected SCRs while standardized betas derived from the former did not differ significantly ($t(65)$ = −0.26, p = 0.794, $d$ = −0.03). Second, standardized betas derived from ranked and non-ranked analyses were comparable for fear ratings ($t(32)$ = 1.26, p = 0.218, $d$ = 0.22) but not for SCRs with significantly higher betas for non-ranked as opposed to ranked SCRs ($t(98)$ = 2.37, p = 0.020, $d$ = 0.24). Third, standardized betas for CS discrimination were significantly lower than for CS+ and CS− for both SCRs (CS+: $t(65)$ = −15.31, p < 0.001, $d$ = −1.88 and CS−: $t(65)$ = −12.34, p < 0.001, $d$ = −1.52) and BOLD fMRI (CS+: $t(11)$ = −4.65, p < 0.001, $d$ = −1.34 and CS−: $t(11)$ = −3.05, p = 0.011, $d$ = −0.88), while for ratings, standardized betas for CS discrimination were higher than for the CS− ($t(21)$ = 3.11, p = 0.005, $d$ = 0.66) and comparable to those for the CS+ ($t(21)$ = −0.57, p = 0.572, $d$ = −0.12). Furthermore standardized betas were larger for the CS+ than for the CS− for SCRs ($t(65)$ = 3.79, p < 0.001, $d$ = 0.47), ratings ($t(21)$ = 3.12, p = 0.005, $d$ = 0.67) and BOLD fMRI ($t(11)$ = 4.34, p = 0.001, $d$ = 1.25). Fourth, standardized betas derived from regressions predicting fear recall were significantly higher than for reinstatement-test for both SCRs ($t(124)$ = 4.35, p < 0.001, $d$ = 0.86) and fear ratings ($t(40)$ = 5.15, p < 0.001, $d$ = 1.76).

## Discussion

In fear conditioning research, little is known about longitudinal reliability (in the literature often referred to as test–retest reliability) for common outcome measures and almost nothing is known about their internal consistency and to what extent predictability across experimental phases is possible.

Here, we aimed to fill this gap and complement traditionally used approaches focusing on ICCs (summarized in *Supplementary file 1*) with (1) analyses of response similarity, (2) the degree of overlap of individual-level brain activation patterns as well as (3) by exploring longitudinal reliability at the group level in addition to (4) internal consistency across outcome measures.

Moreover, we also directly investigated predictability of responding from one experimental phase to subsequent experimental phases. For all analyses, we followed a multiverse-inspired approach (*Parsons, 2020*) by taking into account different data specifications.

Overall, longitudinal group-level reliability was robust for SCRs (see *Figure 4*) and the BOLD response (see *Table 2B*) while longitudinal individual-level reliability as assessed by ICCs (see *Figure 1C–F*), and individual-level BOLD activation overlap (see *Table 2A*) was more limited across outcome measures and data specifications – particularly during extinction training. This is in line with previous work in fear conditioning (*Cooper et al., 2022b*; *Fredrikson et al., 1993*; *Ridderbusch et al., 2021*; *Torrents-Rodas et al., 2014*; *Zeidan et al., 2012*) reporting figures for longitudinal individual-level reliability comparable to ours across outcome measures (SCRs, fear ratings, BOLD fMRI) and experimental phases. Importantly, however, it remains a challenge to interpret the results as benchmarks for ICCs are derived from psychometric work on trait self-report measures and it is plausible that what is interpreted as 'low' and 'high' reliability in experimental work should be substantially lower (*Parsons et al., 2019*).

Our complementary analyses beyond traditional ICCs indicate that SCRs of one individual at T0 were not more similar to responses of the same individual at T1 than compared to others at T1 (see *Figure 2*). For BOLD fMRI, however, acquisition-related individual BOLD activation patterns at T0 were more similar to their own activation patterns at T1 than to other individuals' activation patterns (see *Figure 3*). This was, however, not the case for extinction. Hence, this may suggest that BOLD fMRI might be more sensitive to detect similarity at individual-level responses within participants than SCRs in our data – maybe due to the dependence on spatial (i.e., voxel-by-voxel) rather than temporal (i.e., trial-by-trial) patterns.

Furthermore, we observed a few differences in longitudinal reliability at the individual level depending on data processing specifications (see also *Parsons, 2020*). For most data specifications, reliability was slightly higher for log-transformed and range-corrected SCRs (as opposed to raw and only log-transformed data) while – in contrast to what has been shown for other paradigms and outcome measures (*Baker et al., 2021*; see also https://shiny.york.ac.uk/powercontours/) – an increasing number of trials included in the calculation of ICCs did not generally improve reliability (see *Figure 1—figure supplements 3–8*). Together, this suggests that longitudinal reliability at the individual level is relatively stable across different data transformations and paradigm specifications (e.g., number of trials within the range used here, i.e., 1 to maximum 14) which is important information facilitating the integration of previous work using different time intervals, reliability indices, and paradigms (see *Supplementary file 1*; *Cooper et al., 2022b*; *Fredrikson et al., 1993*; *Ridderbusch et al., 2021*; *Torrents-Rodas et al., 2014*; *Zeidan et al., 2012*).

In contrast, we observed quite robust longitudinal reliability at the group level for both SCRs (see *Figure 4*) and BOLD fMRI (see *Table 2B*) between both time points with substantial (i.e., up to 90%) overlap in group-level BOLD fMRI activation patterns (whole brain and ROI based) as well as substantial (i.e., up to 76%) explained variance at T1 by variance at T0 for SCRs. However, this was generally only true for acquisition but not extinction training. This pattern of higher reliability during acquisition compared to extinction training has been described in the literature (SCRs: *Fredrikson et al., 1993*; *Zeidan et al., 2012*) and was also evident in the similarity analyses of BOLD fMRI and the group-level reliability of SCRs. While this pattern did not emerge across all analyses, it appears to be particularly present when examining reliability of CS discrimination as it was the case for BOLD fMRI and as it also emerged in individual-level reliability analyses of CS discrimination in SCRs (internal consistency and ICCs) and fear ratings (ICCs). Since CS discrimination is typically lower during extinction as compared to acquisition training, this restriction of variance potentially resulted in a floor effect which might have lowered the internal consistency and longitudinal reliability of CS discrimination during extinction training.

Reports regarding this discrepancy between group- and individual-level longitudinal reliability were recently highlighted for a number of (classic) experimental paradigms (*Fröhner et al., 2019*; *Hedge et al., 2018*; *Herting et al., 2018*; *Plichta et al., 2012*; *Schümann et al., 2020*). Our results add fear conditioning and extinction as assessed by SCRs and BOLD fMRI to this list and have important implications for translational questions aiming for individual-level predictions – particularly since findings obtained at the group level are not necessarily representative for any individual within the group (*Fisher et al., 2018*).

In addition to these methods-focused insights, we observed significant associations between responding in different experimental phases for SCR (see *Figure 5A*), fear ratings (see *Figure 5B*) and

BOLD fMRI (see *Figure 6*) revealing that higher responses in previous phases were generally modestly associated with higher responses in subsequent phases in all outcome measures. However, a remarkable amount of predictions were non-significant – which was particularly true for CS discrimination in SCRs and BOLD fMRI. This may be explained by difference scores (i.e., CS+ minus CS−) being generally less reliable (*Infantolino et al., 2018*; *Lynam et al., 2006*) due to a subtraction of meaningful variance (*Moriarity and Alloy, 2021*) particularly in highly correlated predictors (*Thomas and Zumbo, 2012*). Especially at the end of the extinction, CS discrimination is low and hence, variance limited. Therefore, floor effects may contribute to the non-significant effects for extinction end-point performance.

Mixed findings in the literature support both the independence of conditioned responding in different experimental phases (*Bouton et al., 2006*; *Plendl and Wotjak, 2010*; *Prenoveau et al., 2013*; *Shumake et al., 2014*) but also their dependence – particularly in clinical samples (*Foa et al., 1983*; *Rauch et al., 2004*; *Rothbaum et al., 2014*; *Smits et al., 2013a*; *Smits et al., 2013b*). These diverging findings in experimental and clinical studies might point toward a translational gap. However, our work may suggest that the strengths of associations between responding in different phases depended on the specific outcome measure and its specifications (e.g., responses specified as CS discrimination, CS+, or CS−). Yet another explanation – in particular for predictions spanning a 24 hrs delay in experimental phases – might be that individual differences in consolidation efficacy (e.g., how efficiently the fear and extinction memories are consolidated after performing acquisition and extinction training, respectively) may underlie differences in predictability. For example, the performance during a retention or RoF test phase is considered to be determined by the strength of the fear and extinction memory, respectively. Memory strength, however, is not only determined by the strength of the initially acquired memory but also by its consolidation (discussed in *Lonsdorf et al., 2019b*). Thus, as acquisition training preceded the extinction training and reinstatement-test by 24 hrs, it is highly likely that individual differences in consolidation efficacy also impact on performance at test. This has also implications for the common practice of correcting responses during one experimental phase for responding during preceding experimental phases (discussed in *Lonsdorf et al., 2019b*).

Importantly, together with our observation of robust internal consistency (see *Figure 1* and also *Fredrikson et al., 1993*), this pattern of findings suggests that individual-level predictions at short intervals are plausible but might be more problematic for longer time periods as suggested by the limited stability over time in our data.

Yet, we would like to point out that the values we report may in fact point toward good and not limited longitudinal individual-level reliability as our interpretation is guided by benchmarks that were not developed for experimental data but from psychometric work on trait self-report measures. We acknowledge that the upper bound of maximally observable reliability may differ between both cases of application as empirical neuroscientific research inherently comes with more noise. The problem remains that predictions in fear conditioning paradigms appear to not be meaningful for longer periods of time (~6 months). Thus, a key contribution of our work is that it highlights the need to pay more attention to measurement properties in translational research in general and fear conditioning research specifically (e.g., implement reliability calculations routinely in future studies). To date, it remains an open question what 'good reliability' in experimental neuroscientific work actually means (*Parsons et al., 2019*).

Yet, before discussing implications of our results in detail, some reflections on potential (methodological) reasons for (1) limited individual-level but robust group-level reliability and (2) on the role of time interval lengths deserve attention:

First, the limited longitudinal individual-level reliability might indicate that the fear conditioning paradigm employed here – which is a rather strong paradigm with 100% reinforcement rate – may be better suited for investigations of group effects and to a lesser extent for individual difference questions – potentially due to limited variance between individuals (*Hedge et al., 2018*; *Parsons, 2020*; *Parsons et al., 2019*). However, high reliability appears to be possible in principle, as we can conclude from the robust internal consistency of SCRs that we observed. This speaks against a limited between-subject variance and a general impracticability of the paradigm for individual difference research. Hence, we call for caution and warn against concluding from our report that fear conditioning and our outcome measures (SCRs, BOLD fMRI) are unreliable at the individual level.

Second, limited individual-level but robust group-level longitudinal reliability might be (in part) due to different averaging procedures which impacts error variance (*Kennedy et al., 2021*). More precisely, compared to individual-level data, group-level data are based on highly aggregated data resulting in generally reduced error variance which increases group-level reliability.

Third, different operationalizations of the same measurement might have different reliabilities (*Kragel et al., 2021*). For instance, amygdala habituation has been shown to be a more reliable measure than average amygdala activation (*Plichta et al., 2014*) and more advanced analytical approaches such as intraindividual neural response variability (*Månsson et al., 2021*) and multivariate imaging techniques *Kragel et al., 2021*; *Marek et al., 2020*; *Noble et al., 2021*; *Visser et al., 2021* have been suggested to have better (longitudinal) reliability than more traditional analyses approaches. Similarly, methodological advances (e.g., techniques to adjust the functional organization of the brain across participants, *Kong et al., 2021*; or hyperalignment, *Feilong et al., 2021*) in measurement quality and tools may ultimately result in better reliability estimates (*DeYoung et al., 2022*).

Fourth, as discussed above, caution is warranted as traditional benchmarks for 'good' reliability were not developed for experimental work but mainly from psychometric work on trait self-report measures (see above).

Finally, longitudinal reliability refers to measurements obtained under the same conditions and hence it is both plausible and well established that higher reliability is observed at short test–retest intervals (see also *Noble et al., 2021*; *Werner et al., 2022*). Longer intervals are more susceptible to true changes of the measurand – for instance due to environmental influences such as seasonality, temperature, hormonal status, or life events (see *Specht et al., 2011*; *Vaidya et al., 2002*). Indeed most longitudinal reliability studies in the fMRI field used shorter intervals (<6 months, see *Elliott et al., 2020*; *Noble et al., 2021*) than our 6-month interval and hence our results should be conceptualized as longitudinal stability rather than a genuine test–retest reliability. The satisfactory internal consistency speaks against excessive noisiness inherent to our measures as a strong noisiness would also be evident in measurements within one time point and not only emerge across our retest interval. Thus, we rather suggest a true change of the measurand during our retest interval and hence a potentially stronger state than trait dependency.

What do our findings imply? Fear conditioning research has been highlighted as a particularly promising paradigm for the translation of neuroscientific findings into the clinics (*Anderson and Insel, 2006*; *Cooper et al., 2022a*; *Fullana et al., 2020*; *Milad and Quirk, 2012*) and some of the most pressing translational questions are based on individual-level predictions such as predicting treatment success. Our results, however, suggest that measurement reliability may allow for individual-level predictions for (very) short but potentially less so for longer time intervals (such as our 6 months retest interval). Importantly, however, robust group-level reliability appears to allow for group-level predictions over longer time intervals. This applies to SCRs and BOLD fMRI in our data but note that the latter was not investigated for fear ratings. A potential solution and promising future avenue to make use of both good group-level reliability and individual-level predictions might be the use of homogenous (latent) subgroups characterized by similar response profiles (e.g., rapid, slow or no extinction, *Galatzer-Levy et al., 2013a*) – to exploit the fact that reliability appears to be higher for more homogenous samples (*Gulliksen, 1950*).

While general recommendations and helpful discussions on the link between reliability and number of trials (*Baker et al., 2021*), statistical power (*Parsons, 2020*), maximally observable correlations (*Parsons, 2020*), sample and effect size (*Hedge et al., 2018*; *Parsons, 2020*) considerations exist, our results highlight the need for field and subdiscipline specific considerations. Our work allows for some initial recommendations and insights. First, we highlight the value of using multiple, more nuanced measures of reliability beyond traditional ICCs (i.e,. similarity, overlap, *Fröhner et al., 2019*) and second, the relation between number of trials and reliability in an experiment with a learning component (i.e., no increase in reliability with an increasing number of trials). Importantly, our work can also be understood as an empirically based call for action, since more work is needed to allow for clear-cut recommendations, and as a starting point to develop and refine comprehensive guidelines in the future. We also echo the cautionary note of Parsons that 'estimates of reliability refer to the measurement obtained – in a specific sample and under particular circumstances, including the task parameters' (cf. *Parsons, 2020*). Hence, it is important to remember that reliability is a property of a measure that is not fixed and may vary depending on

task specifications and samples. In other words, reliability is not a fixed property of the task itself, here fear conditioning.

We argue that we may need to take a (number of) step(s) back and develop paradigms and data processing pipelines explicitly tailored to individual difference research (i.e., correlation) or experimental (i.e., group level) research questions (e.g., *Parsons, 2020*) and focus more strongly on measurement reliability in experimental work – which has major consequences on effect sizes and statistical power (*Elliott et al., 2020*). More precisely, multiverse-type investigations (*Parsons, 2020*; *Steegen et al., 2016*) that systematically scrutinize the impact of several alternative and equally justifiable processing and analytical decisions in a single dataset (*Kuhn et al., 2022*; *Lonsdorf et al., 2022*; *Sjouwerman et al., 2022*) – as also done here for transformations and number of trials – may be helpful to ultimately achieve this overarching aim. This could be complemented by systematically varying design specifications (*Harder, 2020*) which are extensively heterogeneous in fear conditioning research (*Lonsdorf et al., 2017a*). Calibration approaches, as recently suggested *Bach et al., 2020* follow a similar aim.

Such work on measurement questions should be included in cognitive-experimental work as a standard practice (*Parsons, 2020*) and can (often) be explored in a cost and resource effective way in existing data which in the best case are openly available – which, however, requires cross-lab data sharing and data management homogenization plans. Devoting resources and funds to measurement optimization is a valuable investment into the prospect of this field contributing to improved mental health (*Moriarity and Alloy, 2021*) and to resume the path to successful translation from neuroscience discoveries into clinical applications.

# Materials and methods

## Pre-registration

This project has been pre-registered on the Open Science Framework (OSF) (August 03, 2020; retrieved from https://doi.org/10.17605/OSF.IO/NH24G). Deviations from the pre-registered protocol are made explicit in brief in the methods section and reasons are specified in *Supplementary file 2* as recommended by *Nosek et al., 2018*, who note that such deviations are common and occur even in the most predictable analysis plans.

## Participants

Participants were selected from a large cohort providing participants for subsequent studies as part of the Collaborative Research Center CRC 58. Participants from this sample were recruited for this study through a phone interview. Only healthy individuals between 18 and 50 years of age without a history of childhood trauma according to the Childhood Trauma Questionnaire (CTQ, critical cutoffs as identified by *Bernstein et al., 2003*; *Häuser et al., 2011*). Additional exclusion criteria were claustrophobia, cardiac pacemaker, non-MR-compatible metal implants, brain surgery, left handedness, participation in pharmacological studies within the past 2 weeks, medication except for oral contraceptives, internal medical disorders, chronic pain, neurological disorders, psychiatric disorders, metabolic disorders, acute infections, complications with anesthesia in the past and pregnancy. Participants were right handed and had normal or corrected to normal vision. All participants gave written informed consent to the protocol which was approved by the local ethics committee (PV 5157, Ethics Committee of the General Medical Council Hamburg). The study was conducted in accordance with the Declaration of Helsinki. All participants were naive to the experimental setup and received a financial compensation of 170€ for completion of experiments at both time points (T0 and T1).

The total sample consisted of 120 participants (female$_N$ = 79, male$_N$ = 41, age$_M$ = 24.46, age$_{SD}$ = 3.73, age$_{range}$ = 18–34). At T0 on days 1 and 2, in total 13 participants were excluded due to technical issues (day 1: $N$ = 0; day 2: $N$ = 3), deviating protocols (day 1: $N$ = 2; day 2: $N$ = 0) and SCR non-responding (day 1: $N$ = 3; day 2: $N$ = 5, see below for definition of 'non-responding'). Accordingly, the final dataset for the cross-sectional analysis of T0 data consists of 107 subjects (female$_N$ = 70, male$_N$ = 37, age$_M$ = 24.30, age$_{SD}$ = 3.68, age$_{range}$ = 18–34). 84.11% of these participants were aware and 6.54% were unaware of CS–US contingencies. The remaining 9.35% subjects uncertain of the CS–US contingencies were classified as semi-aware. CS–US contingency awareness of participants was assessed with a standardized post-experimental awareness interview (adapted from *Bechara et al., 1995*). On

average, the US aversiveness was rated on day 1 with a value of 19.82 (SD = 3.28) and on day 2 with a value of 16.46 (SD = 4.75) on a visual analog scale (VAS) ranging from 0 to 25. The US intensity was 8.04 mA (SD = 8.28) on average. Averaged STAI-S (Strait-Trait Anxiety Inventory – State; *Spielberger, 1983*) scores were 35.38 (SD = 5.26) on day 1 and 35.57 (SD = 6.69) on day 2.

At T1, 16 subjects were excluded due to technical issues (day 1: *N* = 1; day 2: *N* = 1), deviating protocols (day 1: *N* = 3; day 2: *N* = 0) and SCR non-responding (day 1: *N* = 5; day 2: *N* = 6; see below for definition of 'non-responding'). Additionally, 20 participants dropped out between T0 and T1 leaving 71 subjects for longitudinal analyses (female$_N$ = 41, male$_N$ = 30, age$_M$ = 24.63, age$_{SD}$ = 3.77, age$_{range}$ = 18–32). 88.73% of the participants were aware and 1.41% were unaware of CS–US contingencies. The remaining 9.86% were classified as semi-aware. US aversiveness was rated with *M* = 19.96 (SD = 2.99) on day 1 and with *M* = 17.73 (SD = 3.90) on day 2 (VAS = 0–25). On average, the US intensity amounted to 9.76 mA (SD = 13.18). Averaged STAI-S scores were 36.33 (SD = 6.09) on day 1 and 35.83 (SD = 7.10) on day 2.

## Experimental design

Here, we reanalyzed pre-existing data that are part of a larger longitudinal study that spanned six time points. In the current study, we included data from a 2-day fear conditioning experiment which were collected at two time points (T0 and T1) 6 months apart. The 2-day experimental procedure and the stimuli were identical at both time points. Measures acquired during the full longitudinal study that are not relevant for the current work such as questionnaires, hair, and salivary cortisol are not described in detail here. For an illustration of the experimental design, see also *Figure 7*.

## Experimental protocol and stimuli

The protocol consisted of a habituation and a fear acquisition training phase on day 1 and an extinction training, reinstatement, and reinstatement-test phase on day 2. Acquisition and extinction training included 28 trials each (14 CS+/14 CS−), habituation and the reinstatement-test phase 14 trials each (7 CS+/7 CS−). Acquisition training was designed as delay conditioning with the US being presented 0.2 s before CS+ offset with 100% reinforcement rate (i.e., all CS+ presentations followed by the US). CSs were two light gray fractals (RGB [230, 230, 230]), 492*492 pixels presented in a pseudo-randomized order, with no more than two identical stimuli in a row, for 6–8 s (mean: 7 s). During the intertrial interval (ITI), a white fixation cross was shown for 10–16 s (mean: 13 s). Reinstatement consisted of three trials with a duration of 5 s each presented after a 10 s ITI. Reinstatement USs were delivered 4.8 s after each trial onset. The reinstatement phase was followed by a 13 s ITI before the next CS was presented during reinstatement-test. All stimuli were presented on a gray background (RGB [100, 100, 100]) using *Presentation software, 2010* (Version 14.8, Neurobehavioral Systems, Inc, Albany, CA USA) keeping the context constant to avoid renewal effects (*Haaker et al., 2014*). Visual stimuli were identical for all participants, but allocation to CS+/CS− and CS type of the first trial of each phase were counterbalanced across participants.

The electrotactile US consisted of a train of three 2 ms electrotactile rectangular pulses with an interpulse interval of 50 ms generated by a Digitimer DS7A constant current stimulator (Welwyn Garden City, Hertfordshire, UK) and was administered to the back of the right hand of the participants through a 1-cm diameter platinum pin surface electrode. The electrode was attached between the metacarpal bones of the index and middle finger. The US was individually calibrated in a standardized stepwise procedure controlled by the experimenter aiming at an unpleasant, but still tolerable level rated by the participants between 7 and 8 on scale from zero (=stimulus was not unpleasant at all) to 10 (=stimulus was the worst one could imagine within the study context). Participants were, however, not informed that we aimed at a score of 7–8.

## Outcome measures
### Skin conductance responses

SCRs were acquired continuously during each phase of conditioning using a BIOPAC MP 100 amplifier (BIOPAC Systems, Inc, Goleta, CA, USA) and Spike 2 software (Cambridge Electronic Design, Cambridge, UK). For analog to digital conversion, a CED2502-SA was used. Two self-adhesive hydrogel Ag/AgCl-sensor recording SCR electrodes (diameter = 55 mm) were attached on the palm of the left hand on the distal and proximal hypothenar. A 10 Hz lowpass filter and a gain of 5Ω were

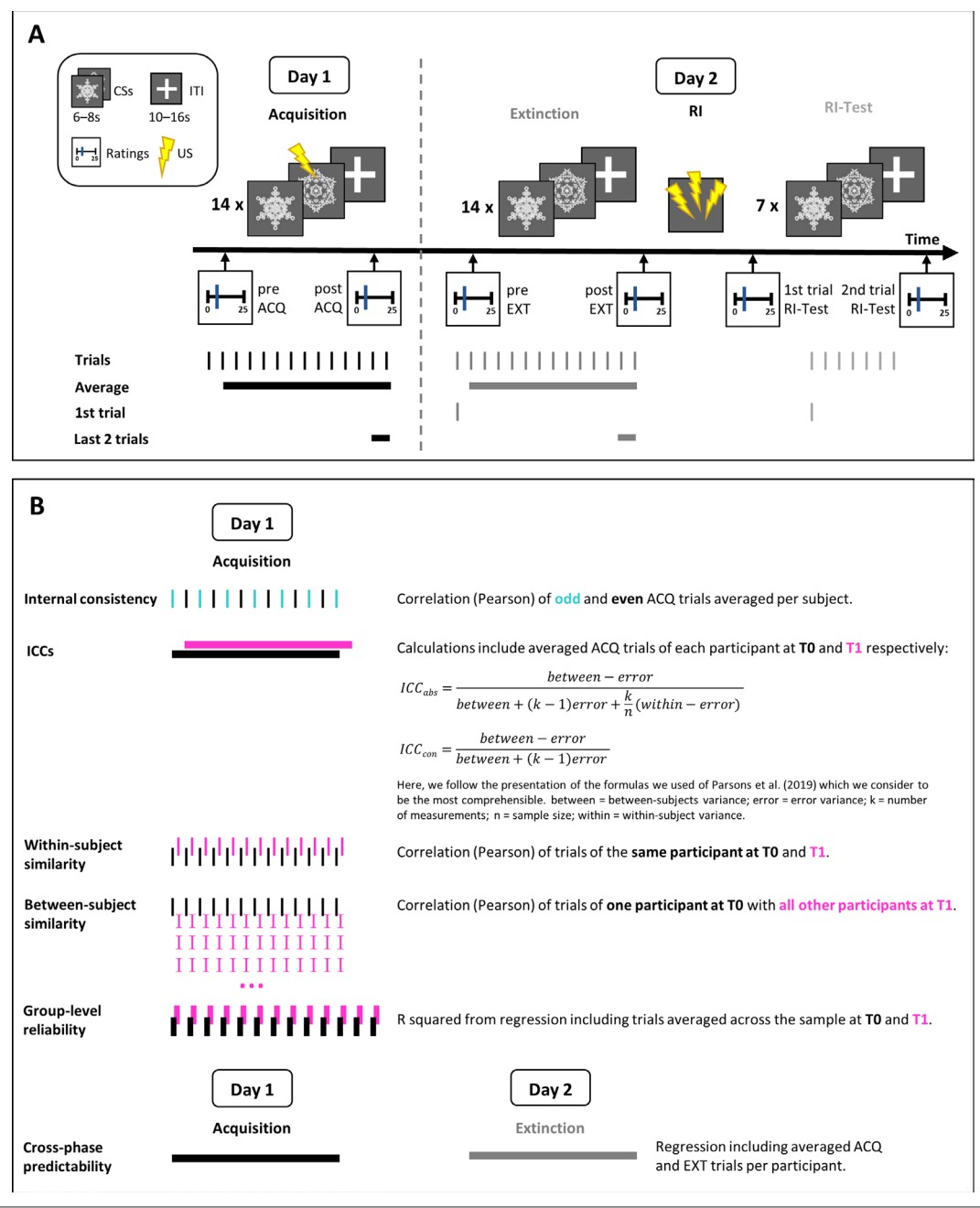

**Figure 7.** Illustration of the experimental design (**A**) and of the calculations of different measures for skin conductance responses (SCRs) including (averaged) acquisition trials (**B**). Note that the habituation phase is not shown in the figure, but described in the text.

applied. Data were recorded at 1000 Hz and later downsampled to 10 Hz. Subsequently, SCRs were scored semi-manually using the custom-made computer program EDA View (developed by Prof. Dr. Matthias Gamer, University of Würzburg). The program is used to quantify the SCR amplitude based on the trough-to-peak method with the trough occurring at 0.9–3.5 s after CS onset and 0.9–2.5 s after US onset (**Boucsein et al., 2012**; **Sjouwerman and Lonsdorf, 2019**). The maximum rise time was set to maximally 5 s (**Boucsein et al., 2012**) unless the US occurred earlier. SCRs confounded by recording artifacts due to technical reasons, such as electrode detachment or responses moving beyond the sampling window, were discarded and scored as missing values. SCRs smaller than 0.01

µS within the defined time window were defined as zero responses. Participants with zero responses to the US in more than two-thirds (i.e., more than 9 out of 14) of US acquisition trials were classified as non-responders on day 1. On day 2, non-responding was defined as no response to any of the three reinstatement USs.

SCR data were prepared for response quantification by using *MATLAB, 2016* version R2016b. No learning could have possibly taken place during the first CS presentations as the US occurred only after the CS presentation. Consequently, the first CS+ and CS− trials during acquisition training were excluded from analyses. Hence, a total of 26 trials (13 differential SCRs) for the acquisition training phase were included in the analyses. For US analyses, all 14 trials were entered into the analyses.

Similarly, responses to the first CS+ and CS− during extinction training have to be considered a 24 hrs delayed test of fear recall as no extinction learning could have taken place. Hence, the first trial and the remaining trials of the extinction were analyzed separately. CS discrimination was computed by subtracting (averaged) CS− responses from (averaged) CS+ responses.

## Fear ratings

Fear ratings to the CSs were collected prior to and after acquisition and extinction training as well as after the reinstatement-test. Participants were asked 'how much stress, fear and tension' they experienced when they last saw the CS+ and CS−. After reinstatement-test, ratings referred to (1) the first CS presentation per CS type directly after reinstatement as well as (2) the last CS presentation during reinstatement-test. After acquisition training and the reinstatement-test, subjects were also asked how uncomfortable they experienced the US itself. All ratings were given on a VAS ranging from zero (answer = none) to 100 (answer = maximum). For analyses, the rating scale was reduced to 0–25. Participants had to confirm the ratings via button press. A lack of confirmation resulted in exclusion of the trial from analyses. CS discrimination was computed by subtracting CS− from CS+ ratings.

## BOLD fMRI: data acquisition, preprocessing, and first-level analysis

The inclusion of BOLD fMRI data was not initially planned and is included here as an additional non-pre-registered outcome measure.

### Data acquisition

Functional data were acquired with a 3 Tesla PRISMA whole body scanner (Siemens Medical Solutions, Erlangen, Germany) using a 64-channel head coil and an echo planar imaging sequence (repetition time: 1980 ms, echo time: 30 ms, number of slices: 54, slice thickness: 1.7 mm [1 mm gap], field of view = 132 × 132 mm). T1-weighted structural images were acquired using a magnetization prepared rapid gradient echo (MPRAGE) sequence (TR: 2300 ms, TE: 2.98 ms, number of slices: 240, slice thickness: 1 mm, field of view = 192 × 256 mm).

### Preprocessing

fMRI data analysis was performed using SPM12 (Wellcome Department of Neuroimaging, London, UK) and *MATLAB, 2019*. Preprocessing included realignment, coregistration, normalization to a group-specific DARTEL template and smoothing (6 mm full width at half maximum, FWHM).

### First-level analysis

Regressors for the first-level analysis of acquisition training data included separate regressors for the first CS+ and CS− trials and the remaining CS+ and CS− trials because no learning could have occurred at the first presentation of the CSs. Nuisance regressors included habituation trials, US presentation, fear ratings and motion parameters. Likewise, separate regressors for the first CS+ and CS− trials of extinction (because no extinction has taken place yet) as well as the remaining CS+ and CS− trials were included as regressors of interest in the first-level analysis of extinction data acquired on day 2, while US, rating onset and motion parameters were included as regressors of no interest. No second-level analysis was completed in the current study, instead different analyses were carried out based on first-level models as further detailed in the statistical analysis section.

## Regions of interest

A total of 11 ROIs (i.e., bilateral anterior insula, amygdala, hippocampus, caudate nucleus, putamen, pallidum, NAcc, thalamus, dACC, dlPFC, and vmPFC) were included in the current study. Amygdala, hippocampus, caudate nucleus, putamen, pallidum, ventral striatum (i.e., NAcc), and thalamus anatomical masks were extracted from the Harvard-Oxford atlas (*Desikan et al., 2006*) at a maximum probability threshold of 0.5. The anterior insula was defined as the overlap between the thresholded anatomical mask from the Harvard-Oxford atlas (threshold: 0.5) and a box of size 60 × 30 × 60 mm centered around MNI$xyz$ = 0, 30, 0 based on anatomical subdivisions (*Nieuwenhuys, 2012*). The cortical ROI dlPFC and dACC were created by building a box of size 20 × 16 × 16 mm around peak voxels obtained in a meta-analysis (with the $x$ coordinate set to 0 for the dACC) (left dlPFC: MNI$xyz$ = −36, 44, 22, right dlPFC: MNI$xyz$ = 34, 44, 32, dACC: MNI$xyz$ = 0, 18, 42, *Fullana et al., 2016*). As previously reported (*Lonsdorf et al., 2014*), the cortical ROI vmPFC was created by using a box of size 20 × 16 × 16 mm centered on peak coordinates identified in prior studies of fear learning (vmPFC: MNI$xyz$ = 0, 40, −12, e.g., *Kalisch et al., 2006*, *Milad et al., 2007*) with the x coordinate set to 0 to obtain masks symmetric around the midline.

All analyses of BOLD fMRI as described below were conducted separately not only for the whole brain but also for these 11 selected ROIs.

## Statistical analyses

For a comprehensive overview of which analysis was carried out for which outcome measures, stimuli, phases and data transformations (see *Table 1*). For an illustration of which data were included in the different analyses, see also *Figure 7B*.

### Internal consistency

We assessed the internal consistency of SCRs for both time points and experimental phases separately (for details, see *Table 1*): trials of the respective time point and phase were split into odd and even trials (i.e., odd–even approach) and averaged for each individual subject. Averaged odd and even trials were then correlated by using Pearson's correlation coefficient. To obtain a rather conservative result, we refrained from applying the Spearman–Brown prophecy formula. We considered the odd–even approach as the most appropriate since our paradigm constitutes a learning experiment and we suggest that adjacent trials measure a more similar construct compared to other possible splits of trials such as a split into halves or a large number of random splits as implemented in the permutation-based approach recommended by *Parsons et al., 2019*. Calculations of internal consistency were not possible for fear ratings and BOLD fMRI due to the limited number of data points for fear ratings and an experimental design that did not allow for a trial-by-trial analysis of BOLD fMRI data. Internal consistency was interpreted using benchmarks for unacceptable (<0.5), poor (>0.5 but <0.6), questionable (>0.6 but <0.7), acceptable (>0.7 but <0.8), good (>0.8 but <0.9), and excellent (≥0.9) (*Kline, 2013*).

## Longitudinal reliability at the individual and group level

While internal consistency indicates the extent to which all items of a test or – here, trials of an experimental phase – measure the same construct (*Revelle, 1979*), longitudinal reliability reflects the variability across two or more measurements of the same individual under the same conditions and is therefore indicative of the degree of correlation and agreement between measurements (*Koo and Li, 2016*). For calculations of longitudinal reliability, we included data from both time points T0 and T1 from the same experimental phase. To capture different aspects of longitudinal reliability, we chose a dual approach of calculating longitudinal reliability at both (1) the individual level and (2) at the group level (for details see also *Table 1*). To this end, longitudinal reliability at the individual and group level indicates to which extent responses within the same individual and within the group as a whole are stable over time. More precisely, whereas longitudinal reliability at the individual level takes into account the individual responses of participants, which are then related across time points, reliability at the group level first averages the individual responses across the group and then relates them across time points. Reliability at the individual level inherently includes the group level, as it is calculated for the sample as whole, but the individual responses are central to the calculation. Contrarily, for reliability at the group level, the calculation is carried out using group averages.

Reliability at the individual level was investigated as (1) ICCs encompassing both time points, (2) within- and between-subject similarity of individual trial-by-trial responding (i.e., SCRs) or BOLD fMRI activation patterns between time points, and (3) as the degree of overlap of significant voxels between time points within an individual (for methodological details see below). Reliability at the group level was investigated as (1) trial-by-trial group average SCRs and (2) the degree of overlap of significant voxels between time points within the group as a whole (for methodological details see below).

Assessments of internal consistency, within- and between-subject similarity, overlap at the individual and group level as well as longitudinal reliability of SCRs at the group level were not pre-registered but are included as they provide valuable additional and complementary information. Overlap and similarity analyses follow the methodological approach of *Fröhner et al., 2019*.

## Longitudinal reliability at the individual level

### Intraclass correlation coefficients

ICCs were determined separately for each experimental phase by including data from both time points T0 and T1. Generally, larger ICCs indicate higher congruency of within-subject responding between time points and increased distinction of subjects from each other (*Noble et al., 2021*). *Parsons et al., 2019* recommend the calculations of ICCs in cognitive-behavioral tasks through a two-way mixed-effects model of single rater type labeled ICC(2,1) (absolute agreement, in the following referred to as $ICC_{abs}$) and ICC(3,1) (consistency, in the following referred to as $ICC_{con}$) according to *Shrout and Fleiss, 1979* convention and to report their 95% CIs. Due to their slightly different calculations, $ICC_{abs}$ tends to be lower than $ICC_{con}$ (see *Table 1*).

However, as the pre-registered mixed-effects approach resulted in non-convergence of some models for SCRs and ratings, we implemented an analysis of variance (ANOVA) instead of the mixed-effects approach to calculate $ICC_{abs}$ and $ICC_{con}$ (*Shrout and Fleiss, 1979*). To calculate ICCs for BOLD fMRI (additional not pre-registered analyses), the SPM-based toolbox fmreli (*Fröhner et al., 2019*) was used. BOLD fMRI ICCs were determined for each voxel and averaged across the whole brain and for selected ROIs.

Furthermore, we investigated whether or to what extent ICCs change when ICC calculations were based on different numbers of trials. To this end, we included (additional non-pre-registered) analyses of trial-by-trial ICCs for SCRs in the supplementary material: First, ICCs were only computed for the first trial. Then, all subsequent trials of the respective phase were added stepwise to this first trial. After each step, trials were averaged and ICCs were calculated (see *Figure 1—figure supplements 3–8*).

Within the figures, values less than 0.5 are classified as poor reliability, values between 0.5 and 0.75 as indicative of moderate reliability, values between 0.75 and 0.9 are classified as good reliability and values greater than 0.9 as excellent reliability, as suggested by *Koo and Li, 2016*. These benchmarks are included here to provide a frame of reference but we point out that these benchmarks are arbitrary and should hence not be overinterpreted in particular in the context of responding in experimental paradigms as these benchmarks have been developed in different contexts (i.e., trait self-report measures).

### Within- and between-subject similarity

Both ICCs and within-subject similarity indicate to which extent responses of an individual at one time point are comparable to responses of the same individual at a later time point. Both were calculated separately for each experimental phase by including data from both time points. There are, however, two main differences: First, ICCs were calculated by decomposition of variances as applied for ANOVA, whereas similarity was calculated as correlation of responses between both time points (1) within one individual (within-subject similarity) and (2) between this individual and all other individuals (between-subject similarity). Second, while ICCs are interpreted in terms of absolute values using cutoffs that provide information on the quantity of longitudinal reliability, within-subject similarity was compared to between-subject similarity showing if responses of one subject at T0 were more similar to themselves at T1 than to responses of all others at T1. The approach to the assessment of similarity was derived from the idea of representational similarity analysis (RSA) introduced by *Kriegeskorte et al., 2008* and previously used by *Fröhner et al., 2019* for the comparison of fMRI BOLD activation patterns between different sessions.

Here, within-subject similarity was calculated by correlating (Pearson's correlation coefficient) (1) individual trial-by-trial SCRs and (2) the first-level response patterns of brain activation for CS discrimination (i.e., CS+ > CS−) of each individual subject between T0 and T1 resulting in one value of within-subject similarity per subject (e.g., SCR acquisition trials of subject 1 at T0 were correlated with SCR acquisition trials of subject 1 at T1). Between-subject similarity was calculated by correlating trial-by-trial SCRs or the first-level response patterns of brain activation of each individual subject at T0 with those of all other individuals at T1 (e.g., SCR acquisition trials of subject 1 at T0 were correlated with SCR acquisition trials of subject 2–71 at T1). This resulted in 70 correlation coefficients for each subject. These correlation coefficients were then averaged to yield one correlation coefficient per subject as an indicator of between-subject similarity.

For comparisons of within- and between-subject similarity in SCR and BOLD fMRI, similarities were Fisher $r$-to-$z$ transformed and compared using paired $t$-tests or Welsh tests in cases where the assumption of equal variances was not met. Cohen's $d$ is reported as effect size.

Note that within-subject similarities of SCRs could not be calculated for participants with a single non-zero response at the same trial (e.g., trial 1) at both time points or only zero responses to the CS+ or CS− in one particular phase. This is because arrays that include only zeros can not be correlated and correlations of 1 (e.g., resulting from non-zero responses at the same trial at both time points) result in infinite Fisher $r$-to-$z$ transformed correlations. Thus, different numbers of participants had to be included in the analyses of SCRs during acquisition ($N_{CS\ discrimination} = 65$, $N_{CS+} = 62$, $N_{CS−} = 56$, $N_{US} = 71$) and extinction training ($N_{CS\ discrimination} = 45$, $N_{CS+} = 40$, $N_{CS−} = 32$).

## Overlap at the individual level

For BOLD fMRI, overlap in individual subject activation patterns across both time points was calculated as a third indicator of reliability at the individual level. Thus, overlap was determined separately for experimental phases by including data from both time points T0 and T1. To this end, activation maps from first-level contrasts (here CS+ > CS or CS discrimination) were compared such that the degree of overlap of significant voxels at a liberal threshold of $p_{uncorrected} < 0.01$ between T0 and T1 was determined and expressed as the Dice and Jaccard coefficients (*Fröhner et al., 2019*). Both coefficients range from 0 (no overlap) to 1 (perfect overlap), with the Jaccard index being easily interpretable as percent overlap (*Fröhner et al., 2019*). While overlap reflects the degree of voxels activated at both time points, similarity measures (see above) are based on the correlation of activated voxels between time points and can be considered a continuous approach based on CS+ > CS− contrast specific beta values and not thresholded T-maps.

## Longitudinal reliability at the group level

As opposed to longitudinal reliability at the individual level which indicates the stability of individual responses across time points, longitudinal reliability at the group level refers to how stable group average responding is over time. Longitudinal reliability at the group level was calculated separately for experimental phases by including data from both time points T0 and T1.

We define longitudinal reliability at the group level (1) for SCRs as the percentage of explained variance of group averaged trials at T1 by group averaged trials at T0 (i.e., $R$ squared) and (2) for BOLD fMRI as the degree of overlap of group averaged activated voxels between both time points. Different analysis approaches were chosen as SCR and BOLD fMRI data are inherently different measures: trial-by-trial analyses in fMRI require slow-event related designs with long ITIs as well as fixed trial orders and ideally partial reinforcement rate to not confound CS and US responses (*Visser et al., 2016*). Hence, trial-by-trial analyses were not possible given our design and thus overlap at a group level was defined as overlap at voxel rather than at trial level.

For SCRs, simple linear regressions were computed with group averaged SCR trials at T0 as independent and group averaged SCR trials at T1 as dependent variable and R squared was extracted. This was done separately for experimental phases. Although the Pearson's correlation coefficient is often calculated to determine longitudinal reliability, $R$ squared, which like overlap can also be expressed as a percentage, appears closest to the concept of overlap of significant voxels at T0 and T1 as applied to BOLD fMRI data.

For overlap in BOLD fMRI at the group level, the degree of overlap of significant voxels between both time points was determined for aggregated group-level activations instead of single subject-level

activation patterns (see 'Overlap at the individual level') and expressed using the Dice and Jaccard indices as described above.

## Cross-phases predictability of conditioned responding

Simple linear regressions were calculated to assess the predictability of SCRs and fear ratings across experimental phases at T0. During data analysis, inspection of the data revealed heteroscedasticity. Therefore and deviating from the pre-registration, regressions with robust standard errors were calculated by using the HC3 estimator (*Hayes and Cai, 2007*). Two consecutive phases represent the independent and the dependent variable, respectively, with the preceding phase as the independent variable and the following phase as the dependent variable. For SCR and fear ratings, standardized betas as derived from linear regressions are reported. In simple linear regression, as implemented here, standardized betas can be also interpreted as Pearson's correlation coefficients.

For fMRI data, we adopted the cross-phases predictability analysis of SCR and fear ratings by calculating Pearson's correlation coefficients between patterns of voxel activation (i.e., first-level beta maps). Correlations were first calculated at the individual subject level and subsequently averaged.

Standardized betas (resulting from SCR and fear rating regressions) and correlation coefficients (resulting from BOLD fMRI correlational tests) were interpreted as demonstrating weak, moderate, or strong associations between variables with values of <0.4, ≥0.4, and ≥0.7, respectively (*Dancey and Reidy, 2007*). Tables containing regression parameters beyond the standardized betas depicted in *Figure 5A, B* are presented in the Supplement (see *Supplementary file 7*, *Supplementary file 8*).

For SCR and fear rating predictions, we assessed if predictions differ in their strength or direction when they are summarized across certain data specifications (see *Table 1*). For BOLD fMRI, correlation coefficients were pooled across ROIs. *T*-tests or Welch tests in cases where the assumption of equal variances was not met were performed on individual Fisher *r*-to-*z* transformed standardized betas (SCR and fear ratings) or correlation coefficients (BOLD fMRI). We highlight that these analyses can be interpreted as an example for predictive validity (i.e., the extent to which a score on a test predicts a score on a criterion measure). As our aim here is, however, not validation, we use the term cross-phase prediction throughout. (More precisely, we believe that 'cross-phase predictions' in our study cannot be used interchangeably with 'criterion or predictive validity' since our aim was not to validate one experimental phase against the other. Predictive validity in psychometrics is defined as 'the extent to which a score on a scale (or test) predicts scores on some criterion measure' (cf. *Cronbach and Meehl, 1955*). For instance, a cognitive test for job performance would have predictive validity if the observed correlation between the test score and the performance rating by the company were statistically significant. Rather, we investigated whether responses in earlier experimental phases could predict responses in later experimental phases – both of which cannot be expected to 'measure the same thing'.)

For all statistical analyses described above, a level of p < 0.05 (two-sided) was considered significant. Since we were more interested in patterns of results and less in the result of one specific test, it was not necessary to correct for multiple comparisons. Moreover, multiverse approaches, as approximated in our study, are assumed to be insensitive to multiple comparisons (*Lonsdorf et al., 2022*).

For data analyses and visualizations as well as for the creation of the manuscript, we used R (Version 4.1.3; *R Development Core Team, 2020*) and the R-packages *apa* (*Aust and Barth, 2020*; Version 0.3.3; *Gromer, 2020*), *car* (Version 3.0.10; *Fox and Weisberg, 2019*; *Fox et al., 2020*), *carData* (Version 3.0.4; *Fox et al., 2020*), *cowplot* (Version 1.1.1; *Wilke, 2020*), *DescTools* (Version 0.99.42; *Andri mult, 2021*), *dplyr* (Version 1.0.8; *Wickham et al., 2021*), *effsize* (*Torchiano, 2020*), *flextable* (Version 0.6.10; *Gohel, 2021*), *gghalves* (Version 0.1.1; *Tiedemann, 2020*), *ggplot2* (Version 3.3.5; *Wickham, 2016*), *ggpubr* (Version 0.4.0; *Kassambara, 2020*), *ggsignif* (Version 0.6.3; *Constantin and Patil, 2021*), *gridExtra* (Version 2.3; *Auguie, 2017*), *here* (Version 1.0.1; *Müller, 2020*), *kableExtra* (Version 1.3.1; *Zhu, 2020*), *knitr* (Version 1.37; *Xie, 2015*), *lm.beta* (Version 1.5.1; *Behrendt, 2014*), *lmtest* (Version 0.9.38; *Zeileis and Hothorn, 2002*), *officedown* (Version 0.2.4; *Gohel and Ross, 2022*), *papaja* (Version 0.1.0.9997; *Aust and Barth, 2020*), *patchwork* (Version 1.1.0; *Pedersen, 2020*), *psych* (Version 2.0.9; *Revelle, 2020*), *renv* (Version 0.13.2; *Ushey, 2020*), *reshape2* (Version 1.4.4; *Wickham, 2007*), *sandwich* (*Zeileis, 2004*; *Zeileis, 2006*; Version 3.0.1; *Zeileis et al., 2020*), *stringr* (Version 1.4.0; *Wickham, 2019*), *tidyr* (Version 1.2.0; *Wickham, 2020*), *tinylabels* (Version 0.2.3; *Barth, 2022*), and *zoo* (Version 1.8.8; *Zeileis and Grothendieck, 2005*).

## Acknowledgements

The authors would like to thank Claudia Immisch, Janne Nold, Kevin Rozario, and Habiba Schiller for help with data collection and Karoline Rosenkranz for help with data preprocessing, Mario Reutter for methodological discussions and comments on an earlier draft as well as Juliane Tkotz for support with reproducible manuscript writing.

## Additional information

### Funding

| Funder | Grant reference number | Author |
|---|---|---|
| Deutsche Forschungsgemeinschaft | INST 211/633-2 | Tina B Lonsdorf |
| Deutsche Forschungsgemeinschaft | LO 1980/4-1 | Tina B Lonsdorf |
| Deutsche Forschungsgemeinschaft | LO 1980/7-1 | Tina B Lonsdorf |

The funders had no role in study design, data collection, and interpretation, or the decision to submit the work for publication.

### Author contributions

Maren Klingelhöfer-Jens, Conceptualization, Data curation, Software, Formal analysis, Visualization, Methodology, Writing - original draft, Pre-registration of the study; Mana R Ehlers, Conceptualization, Formal analysis, Visualization, Methodology, Writing - original draft; Manuel Kuhn, Data curation, Software, Investigation, Writing – review and editing; Vincent Keyaniyan, Formal analysis, Visualization, Methodology, Writing – review and editing, Pre-registration of the study; Tina B Lonsdorf, Conceptualization, Resources, Supervision, Funding acquisition, Methodology, Writing - original draft, Pre-registration of the study

### Author ORCIDs

Maren Klingelhöfer-Jens (ID) http://orcid.org/0000-0002-5393-7871
Mana R Ehlers (ID) http://orcid.org/0000-0002-1316-3787
Manuel Kuhn (ID) http://orcid.org/0000-0003-2210-9130
Vincent Keyaniyan (ID) http://orcid.org/0000-0002-5674-5197
Tina B Lonsdorf (ID) http://orcid.org/0000-0003-1501-4846

### Ethics

All participants gave written informed consent to the protocol which was approved by the local ethics committee (PV 5157, Ethics Committee of the General Medical Council Hamburg). The study was conducted in accordance with the Declaration of Helsinki.

### Decision letter and Author response

Decision letter https://doi.org/10.7554/eLife.78717.sa1
Author response https://doi.org/10.7554/eLife.78717.sa2

## Additional files

### Supplementary files

- Supplementary file 1. Overview of experimental specifications and results of five previous studies reporting test–retest reliabilities in human fear conditioning research.
- Supplementary file 2. Deviations from pre-registration.
- Supplementary file 3. $ICC_{abs}$ and $ICC_{con}$ for all data specifications of SCRs.
- Supplementary file 4. $ICC_{abs}$ and $ICC_{con}$ for all data specifications of fear ratings.
- Supplementary file 5. $ICC_{abs}$ and $ICC_{con}$ for CS discrimination during fear acquisition (Acq) and extinction training (Ext).

- Supplementary file 6. Paired sample *t*-tests comparing between- and within-subject similarity for whole brain activation pattern as well as activation pattern in the ROIs for acquisition training (Acq) and extinction training (Ext).
- Supplementary file 7. Detailed results of linear regressions: SCR.
- Supplementary file 8. Detailed results of linear regressions: fear ratings.
- Transparent reporting form
- MDAR checklist

## Data availability

The data that support the findings of this study and the R Markdown files that generate this manuscript are openly available in Zenodo at https://doi.org/10.5281/zenodo.7323547.

The following dataset was generated:

| Author(s) | Year | Dataset title | Dataset URL | Database and Identifier |
|---|---|---|---|---|
| Klingelhöfer-Jens M, Ehlers MR, Kuhn M, Keyaniyan V, Lonsdorf TB | 2022 | Robust group- but limited individual-level (longitudinal) reliability and insights into cross-phases response prediction of conditioned fear | https://doi.org/10.5281/zenodo.7323547 | Zenodo, 10.5281/zenodo.7323547 |

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
