## [Editor Report]

The authors assess the psychometric properties of behavioral, psychophysiological, and brain imaging measures of fear conditioning. Six-month retest reliability was generally low, whereas internal-consistency reliability was generally high. At the group level, reliability and criterion validity were generally good. Most measurements proved sensitive to data analytical choices. Results are framed within a larger discussion of the role of measurement properties in individual difference research and clinical translation and have the potential to serve as an important building block towards improvement in both these areas.

---

## [Decision Letter]

**Decision letter after peer review:**

Thank you for submitting your article "Robust group- but limited individual-level (longitudinal) reliability and insights into cross-phases response prediction of conditioned fear" for consideration by *eLife*. Your article has been reviewed by 2 peer reviewers, and the evaluation has been overseen by Drs. Shackman (Reviewing Editor) and Baker (Senior Editor).

The reviewers highlighted several strengths of the manuscript.

– We very much liked the paper and appreciate the potentially important implications for the field.

– This is a very comprehensive and thoughtful effort

– The approach is thorough, with a range of analysis approaches, including within- and between-subjects similarity, the individual-level overlap of fMRI results, ICCs, and cross-sectional reliability. It is important to determine these values so that researchers can discard incorrect assumptions, such as the belief that threat responses at baseline can be predictive of treatment responses in patient populations.

– The conclusions of this work are largely supported by the data and methodological approach, and this is a good benchmark for the field.

– The overall approach is excellent and represents the vanguard of open science practices (preregistration, all materials freely available, documentation of analysis deviations, multiverse analyses, etc.).

– This comprehensive approach drives home the conclusion that specific analytic choices and researcher "degrees of freedom" can have sometimes drastic effects on fundamental measurement properties. I think this underlines what I view as the key contribution of this manuscript: empirically highlighting the need for the fear conditioning field to pay more attention to measurement properties.

– Going beyond standard associative measures of reliability (ICCs) is an important contribution of this work, as they allow the authors to comment on nuances of individual-difference reliability that are not possible with the coarser ICCs. In turn, this facilitates researchers in making more informed decisions regarding the design of fear conditioning tasks to assess individual differences.

– The fMRI results are a particular strength, as fMRI continues to be a common fear conditioning index, yet its measurement properties in this context remain critically understudied. The choice to use standard ICCs in conjunction with similarity approaches is particularly fruitful here, as in conjunction with overlap metrics we now have a much better appraisal of the different components of reliability in fMRI data – and potential explanations for differences between behavioral and fMRI reliabilities.

– The poor reliability identified by several of these approaches is likely to be of great importance to this large, translational field.

Nevertheless, several aspects of the manuscript somewhat diminished enthusiasm, as outlined below.

The reviewers have discussed their critiques with one another and the Reviewing Editor, who has drafted this to help you prepare a revised submission.

Major Revisions:

1. Terminology. In terms of aligning with the psychometric and measurement literature, and to assist in clarity, I suggest the authors consistently use established measurement terms whenever possible. For example, "cross-sectional validity" is more often referred to as internal consistency, and the cross-phase predictions are an example of criterion/predictive validity. As the fear conditioning field is relatively new to formal measurement theory, it would be helpful to have everyone "on the same page" as decades of prior measurement work.

2. Increase Accessibility for a Broad Scientific Audience. We strongly recommend a thorough rewrite of the introduction and discussion. Essentially, the paper needs to be substantially 'dumbed down' to become accessible to the broad readership it deserves. The introduction should start by explaining the rationale for choosing to use fear conditioning methods, why fear conditioning research is so important, and mention some key insights generated by this type of research. (Some of this is included and/or alluded to, but could be clarified further for readers not working with fear conditioning in humans).

3. Clarify the Study Design. In the introduction, clearly explain the study design and the kinds of the data included in the analyses, both in the text and using a diagram with a visual overview of a typical trajectory (conditioning and extinction phases) either at the individual or group level or both. Ideally include example results of all the kinds included here, SCR, ratings and fMRI data and at both individual and group level, to really give the reader a sense of what's gone into in your analyses. Relatedly: The authors should take care to clearly articulate why they chose to use using 1st point, endpoint and average to investigate cross-phase predictability in the manuscript. Also, why it makes sense to look at raw, log-transformed, ranked, non-ranked data etc.

4. Clarify the Significance of Individual Differences and Reliability

a. A bit more explanation for why individual-level measurement is important for addressing clinical individual differences would be appreciated. At present, it is described as important but the "why" is not really fleshed out. For example, content about heterogeneity in clinical presentations requiring individual-level metrics (as opposed to categorical diagnoses) could be useful here.

b. When explaining the importance of reliability and prediction, please try to give concrete examples. Imagine your reader is an eager but inexperienced grad student who might wish to embark upon fear conditioning research, but has no experience and needs guidance at every level. How many trials to include, what type of measures, how many participants, how large can effects be expected to be – the discussion should clarify how your analyses inform these decisions.

5. Clarify the Significance of "Group-Level" Reliability. The authors analyze group-level reliability and situate this as being improved in relation to individual-level reliability. I think the authors' explanation for the importance of group-level reliability is not fully fleshed out and at present, it is not clear what new information the field can take from the reported group-level reliabilities. At present there seems to be some circular logic here: group-level reliability is studied because group-level inferences are important, and they are important because they are studied. I am concerned that non-specialist readers might get the wrong impression from this and conclude that fear conditioning is only useful for group-level inference, as well as continue to perpetuate the paradox described so well in Hedge et al., 2018, and others (e.g. work from Russ Poldrack's group).

6. Internal Consistency Formula. The internal consistency (cross-sectional reliability) calculation used is not well-justified, and potentially needs additional parameters. It is not clear why the authors deviate from the internal consistency calculation described in Parson, Kruijt, and Fox et al., 2019, especially given that these procedures are used for other metrics elsewhere in the manuscript. I request that the authors either use the bias-corrected formula from Parson et al., 2019 or justify the use of the current calculation.

7. ROIs. For the fMRI analyses, the authors use an ROI approach based on prior studies of fear acquisition and extinction. The majority of the most consistently identified regions (as seen in meta-analyses, Fullana et al., 2016, 2018) are analyzed. However, it is not clear why other regions are omitted, particularly given meta-analytic evidence. Striatal regions and the thalamus are the most notable omissions. Further, a weakness is that functional ROIs in this study were based on peak coordinates from a handful of prior studies, instead of meta-analytically identified coordinates. As such, I do not think the authors present the strongest foundation for making conclusions about the reliability of fear conditioning fMRI Data. I request that the authors include additional ROIs for the canonical fear network, or justify why the particular ROIs that are currently reported were the only ones used. I also strongly suggest using meta-analytic coordinates to determine ROIs for these analyses.

8. Reliability and the Need for Nuance

a. The authors structure the manuscript around the premise that reliability is essential in conducting solid individual-differences science, which I agree with wholeheartedly. However, I think the authors rely on relatively arbitrary cut-offs for classifying reliability as good/poor/etc to an extent that is not warranted, particularly in the context of the Discussion, and it takes away from the impact of this effort. As the authors point out, these categorical cut-offs are more guidelines than strict rules, yet the manuscript is structured around the premise that individual-level reliability is problematically poor. Many cut-off recommendations are based on psychometric work on trait self-report measures that usually assume fewer determinants/sources of error than would be seen in neuroscience experiments, which in turn allows for larger ceilings for effect sizes and reliability. The current manuscript does not address this issue and what meaningful (as opposed to good) fear conditioning reliability is when moving away from the categorical cut-offs. In other words, is it possible that the authors actually observed "good" reliability in the context of fear conditioning work, and that this reliability is lower than other types of paradigms is just inherent to the construct being studied?

b. The framing of the manuscript could be adjusted, such that less emphasis is placed on arbitrary cut-off metrics and more about what these reliabilities mean in the context of fear conditioning paradigms. To be clear, I think the authors already address this to a degree in their Discussion, but perhaps need to go a step further and expand on the challenges of establishing reliability in this field and explicitly address why common cut-offs are perhaps not appropriate.

c. It may be more appropriate to use numbers rather than labels/benchmarks in the reporting of results in the Results section, i.e. reporting the r-value instead of "poor to questionable" etc.

9. Clarify Implications and Recommendations. The concrete implications of the research, and recommendations arising from it, should be clearly spelled out in the Abstract and Discussion, for the greatest utility.

---

## [Author Response]

Major Revisions:1. Terminology. In terms of aligning with the psychometric and measurement literature, and to assist in clarity, I suggest the authors consistently use established measurement terms whenever possible. For example, "cross-sectional validity" is more often referred to as internal consistency, and the cross-phase predictions are an example of criterion/predictive validity. As the fear conditioning field is relatively new to formal measurement theory, it would be helpful to have everyone "on the same page" as decades of prior measurement work.

We agree with the importance of homogeneous terminology as outlined by the reviewer/editor. We use the term longitudinal reliability to provide an umbrella term for test-retest reliability with a long test-retest interval, which we compute through different approaches including ICCs, but also with relatively new reliability measures such as similarity and overlap. With this umbrella term we want to increase the understanding of the different measures and help the reader to keep track of them.

The term cross-sectional reliability was used to clearly discriminate this measure from longitudinal reliability. We agree, however, that it may be confusing to introduce a new term for internal consistency. In our revised manuscript, we follow the reviewers’/editors’ suggestion and use internal consistency throughout. We have also moved Table 1 to an earlier position in the manuscript as it assists in clarity by providing an overview of the different reliability types and detailed definitions as well as formulas (see also reviewer comment 11).

However, we do not fully agree with exchanging “cross-phase predictions” with “criterion or predictive validity”, because we do not want to validate one experimental phase against the other. Predictive validity in psychometrics is defined as “the extent to which a score on a scale (or test) predicts scores on some criterion measure” (cf. Cronbach and Meehl, 1955). For instance a cognitive test for job performance would have *predictive validity* if the observed correlation between the test score and the performance rating by the company were statistically significant.

Rather, we investigate whether responses in earlier experimental phases can predict responses in later experimental phases – both of which cannot be expected to “measure the same thing”. This is of relevance, as in the literature it is often assumed that this is true without strong empirical support due to a lack of available studies on this topic and heterogeneous findings in the literature (see e.g. Lonsdorf et al., 2017, Lonsdorf et al., 2020). We hope the reviewer and editor agree with this reasoning.

Cronbach, L.J., and Meehl, P.E. (1955). Construct validity for psychological tests. Psychological Bulletin, 52, 281-302.[1]

Lonsdorf, T. B., Menz, M. M., Andreatta, M., Fullana, M. A., Golkar, A., Haaker, J., … Merz, C. J. (2017). Don’t fear ’fear conditioning’: Methodological considerations for the design and analysis of studies on human fear acquisition, extinction, and return of fear. Neuroscience and Biobehavioral Reviews, 77, 247–285. https://doi.org/10.1016/j.neubiorev.2017.02.026

Lonsdorf, T. B., Merz, C. J., and Fullana, M. A. (2019). Fear Extinction Retention: Is It What We Think It Is? Biological Psychiatry, 85(12), 1074–1082.

2. Increase Accessibility for a Broad Scientific Audience. We strongly recommend a thorough rewrite of the introduction and discussion. Essentially, the paper needs to be substantially 'dumbed down' to become accessible to the broad readership it deserves. The introduction should start by explaining the rationale for choosing to use fear conditioning methods, why fear conditioning research is so important, and mention some key insights generated by this type of research. (Some of this is included and/or alluded to, but could be clarified further for readers not working with fear conditioning in humans).

We thank the reviewer/editor for raising these points and have restructured our introduction as suggested as well as edited the discussion accordingly. We hope that with these edits, we have made our manuscript more accessible to a wider audience. As the changes are excessive we refrain from quotes here.

3. Clarify the Study Design. In the introduction, clearly explain the study design and the kinds of the data included in the analyses, both in the text and using a diagram with a visual overview of a typical trajectory (conditioning and extinction phases) either at the individual or group level or both. Ideally include example results of all the kinds included here, SCR, ratings and fMRI data and at both individual and group level, to really give the reader a sense of what's gone into in your analyses.

We thank the reviewer/editor for pointing out that the study design and how the data went into our analyses were not sufficiently clear. We added a figure to the methods section which illustrates both the design and the calculations of our different measures. We opted for the calculations for the SCRs as an example because this is the outcome for which all calculations were performed.

Relatedly: The authors should take care to clearly articulate why they chose to use using 1st point, endpoint and average to investigate cross-phase predictability in the manuscript. Also, why it makes sense to look at raw, log-transformed, ranked, non-ranked data etc.

We thank the reviewer/editor for the suggestion to justify the specifications of our data even more clearly. We have added some explanations and justifications to the footnotes of Table 1 which we have moved to the end of the introduction.

4. Clarify the Significance of Individual Differences and Reliabilitya. A bit more explanation for why individual-level measurement is important for addressing clinical individual differences would be appreciated. At present, it is described as important but the "why" is not really fleshed out. For example, content about heterogeneity in clinical presentations requiring individual-level metrics (as opposed to categorical diagnoses) could be useful here.

We thank the reviewer/editor for highlighting that this was not sufficiently clear. We added the following paragraph to the introduction:

“Hence, tackling clinical questions regarding individual prediction of symptom development or treatment outcome requires a shift towards and a validation of research methods tailored to individual differences – such as a focus on measurement reliability (Zuo, Xu, and Milham, 2019). This is a necessary precondition when striving for the long-term goal of developing individualized intervention and prevention programs. This relates to the pronounced symptomatic heterogeneity in symptom manifestations between individuals diagnosed with the same disorders (e.g. PTSD, Galatzer-Levy and Bryant, 2013) which is not captured in binary clinical diagnoses as two patients with the diagnosis PTSD may not share a single symptom (Galatzer-Levy and Bryant, 2013).”

b. When explaining the importance of reliability and prediction, please try to give concrete examples. Imagine your reader is an eager but inexperienced grad student who might wish to embark upon fear conditioning research, but has no experience and needs guidance at every level. How many trials to include, what type of measures, how many participants, how large can effects be expected to be – the discussion should clarify how your analyses inform these decisions.

We agree with the reviewer/editor that we have not fully exhausted our possibilities to derive recommendations from our findings. We have added a paragraph to the discussion to make explicit what recommendations can be derived from our work and where to go from here. For many of the questions mentioned in the comment, it is very difficult to provide strong empirically-based recommendations from this single study – even though we wholeheartedly agree and share the desire for clear guidance. Our work should be seen as a starting point to develop and refine such guidelines in the future. For instance, we cannot provide any guidance on how large effect sizes can be expected as this is highly dependent on the specific research question and study sample at stake. We, however, provide recommendations wherever possible and a clear outline for future work. We also refer to our previous methods-focused work for guidance (see e.g., Lonsdorf et al., 2017).

Lonsdorf, T. B., Menz, M. M., Andreatta, M., Fullana, M. A., Golkar, A., Haaker, J., Heitland, I., Hermann, A., Kuhn, M., Kruse, O., Meir Drexler, S., Meulders, A., Nees, F., Pittig, A., Richter, J., Römer, S., Shiban, Y., Schmitz, A., Straube, B., … Merz, C. J. (2017). Don’t fear „fear conditioning“: Methodological considerations for the design and analysis of studies on human fear acquisition, extinction, and return of fear. Neuroscience and Biobehavioral Reviews, 77, 247–285. https://doi.org/10.1016/j.neubiorev.2017.02.026

Example from discussion:

“While general recommendations and helpful discussions on the link between reliability and number of trials (Baker et al., 2021), statistical power (Parsons, 2020), maximally observable correlations (Parsons, 2020), sample and effect size (Hedge et al., 2018; Parsons, 2020) considerations exist, our results highlight the need for field and sub-discipline specific considerations. Our work allows for some initial recommendations and insights. First, we highlight the value of using multiple, more nuanced measures of reliability beyond traditional ICCs (i.e,. similarity, overlap, Fröhner et al. (2019)) and second, the relation between number of trials and reliability in an experiment with a learning component (i.e., no increase in reliability with an increasing number of trials). Importantly, our work can also be understood as an empirically-based call for action, since more work is needed to allow for clear-cut recommendations, and as a starting point to develop and refine comprehensive guidelines in the future.”

5. Clarify the Significance of "Group-Level" Reliability. The authors analyze group-level reliability and situate this as being improved in relation to individual-level reliability. I think the authors' explanation for the importance of group-level reliability is not fully fleshed out and at present, it is not clear what new information the field can take from the reported group-level reliabilities. At present there seems to be some circular logic here: group-level reliability is studied because group-level inferences are important, and they are important because they are studied. I am concerned that non-specialist readers might get the wrong impression from this and conclude that fear conditioning is only useful for group-level inference, as well as continue to perpetuate the paradox described so well in Hedge et al., 2018, and others (e.g. work from Russ Poldrack's group).

We thank the reviewer/editor for pointing out to us that our justification for researching reliability at the group level fell short. We have made further points as to why we think this is important.

Introduction:

“To date, both clinical and experimental research using the fear conditioning paradigm have primarily focused on group-level, basic, general mechanisms such as the effect of experimental manipulations – which is important to investigate (Lonsdorf and Merz, 2017).”

“More precisely, longitudinal reliability at the group level indicates the extent to which responses averaged across the group as a whole are stable over time, which is important to establish when investigating basic, generic principles such as the impact of experimental manipulations. Even though it has to be acknowledged that the group average is not necessarily representative of any individual in the group and the same group average may arise from different and even opposite individual responses at both time points in the same group, group-level reliability is important to establish in addition to individual-level reliability. Group-level reliability is relevant not only to work focusing on the understanding of general, generic processes but also for questions about differences between two groups of individuals such as patients vs. controls (e.g., see meta-analyses of Cooper et al., 2022; Duits et al., 2015). Of note, many fear conditioning paradigms were initially developed to study general group-level processes and to elicit robust group effects (Lonsdorf and Merz, 2017). Hence it is important to investigate both group- and individual-level reliability given the challenges of attempts to employ cognitive tasks that were originally designed to produce robust group effects in individual difference research (Elliott et al., 2020; Hedge et al., 2018; Parsons, 2020; Parsons, Kruijt, and Fox, 2019).”

Discussion:

“First, the limited longitudinal individual-level reliability might indicate that the fear conditioning paradigm employed here – which is a rather strong paradigm with 100% reinforcement rate – may be better suited for investigations of group effects and to a lesser extent for individual difference questions – potentially due to limited variance between individuals (Hedge et al., 2018; Parsons, 2020; Parsons et al., 2019). However, high correlations seem to be possible in principle, as we can conclude from the robust internal consistency of SCRs that we observed. This speaks against a limited between-subject variance and a general impracticability of the paradigm for individual difference research. Hence we call for caution and warn against concluding from our report that fear conditioning and our outcome measures (SCRs, BOLD fMRI) are unreliable at the individual level.”

6. Internal Consistency Formula. The internal consistency (cross-sectional reliability) calculation used is not well-justified, and potentially needs additional parameters. It is not clear why the authors deviate from the internal consistency calculation described in Parson, Kruijt, and Fox et al., 2019, especially given that these procedures are used for other metrics elsewhere in the manuscript. I request that the authors either use the bias-corrected formula from Parson et al., 2019 or justify the use of the current calculation.

We very much appreciate the approach of Parsons et al. (2019) of robust estimation through permutation, in which data are multiple times (Parsons et al. (2019) recommend 5000 times as a minimum) randomly split into two halves, the reliability is estimated for each split and these estimates are averaged. In our case as the fear conditioning paradigm is a learning paradigm, however, we believe this approach cannot be applied without further consideration, since not every random division of the data makes sense. If, for example, the data from the acquisition phase were divided into halves, responses from the beginning and end of this phase would be used to determine reliability. However, because it is a learning experiment, these two halves probably do not measure exactly the same construct. Therefore, we decided to use the odd-even method because we believe that adjacent trials “measure something more similar” than random splits in this specific case (i.e., learning paradigm). We have included a brief justification in the manuscript:

“We considered the odd-even approach as the most appropriate since our paradigm constitutes a learning experiment and we suggest that adjacent trials measure a more similar construct compared to other possible splits of trials such as a split into halves or a large number of random splits as implemented in the permutation-based approach recommended by Parsons et al. (2019).”

7. ROIs. For the fMRI analyses, the authors use an ROI approach based on prior studies of fear acquisition and extinction. The majority of the most consistently identified regions (as seen in meta-analyses, Fullana et al., 2016, 2018) are analyzed. However, it is not clear why other regions are omitted, particularly given meta-analytic evidence. Striatal regions and the thalamus are the most notable omissions. Further, a weakness is that functional ROIs in this study were based on peak coordinates from a handful of prior studies, instead of meta-analytically identified coordinates. As such, I do not think the authors present the strongest foundation for making conclusions about the reliability of fear conditioning fMRI Data. I request that the authors include additional ROIs for the canonical fear network, or justify why the particular ROIs that are currently reported were the only ones used. I also strongly suggest using meta-analytic coordinates to determine ROIs for these analyses.

We thank the reviewer/editor for pointing out that our results and arguments could be strengthened by expanding the number of included ROIs to those identified in previous meta-analyses (Fullana et al., 2016, 2018). To accommodate this suggestion, we have added the caudate nucleus, the putamen, the pallidum, the nucleus accumbens as a proxy for the ventral striatum as well as the thalamus to the list of anatomically defined ROIs. Furthermore, we have created masks for the dACC and the dlPFC based on meta-analytically identified peak coordinates. For the vmPFC, the meta-analysis only identified a non-significant cluster with negative activation for CS+ > CS-, with its peak coordinates more anterior than the activation found in previous publications (see red box in Author response image 1 for our mask and crosshair for meta-analytic peak coordinates) (Kalisch et al., 2006, Milad et al., 2007). Since the involvement of the vmPFC in fear acquisition and especially extinction has still been shown in many publications, we wanted to include the ROI but based its location on previous studies.

**Author response image 1. sa2fig1:** 

Please note, that the general pattern of results for all our measures of reliability has not changed with the addition of regions of interest and the masks based on peak coordinates identified in meta-analyses.An updated detailed methodological description can be found in the methods section (see pages 64 – 65) under ‘Regions of Interest’. The excerpt is also pasted below for your convenience.

“A total of 11 regions of interest (ROIs; i.e., bilateral anterior insula, amygdala, hippocampus, caudate nucleus, putamen, pallidum, nucleus accumbens [NAcc], thalamus, dorsal anterior cingulate cortex [dACC], dorsolateral prefrontal cortex [dlPFC] and ventromedial prefrontal cortex [vmPFC]) were included in the current study. Amygdala, hippocampus, caudate nucleus, putamen, pallidum, ventral striatum (i.e., nucleus accumbens) and thalamus anatomical masks were extracted from the Harvard-Oxford atlas (Desikan et al., 2006) at a maximum probability threshold of 0.5. The anterior insula was defined as the overlap between the thresholded anatomical mask from the Harvard Oxford atlas (threshold: 0.5) and a box of size 60 x 30 x 60 mm centered around MNIxyz = 0, 30, 0 based on anatomical subdivisions (Nieuwenhuys, 2012). The cortical ROI dlPFC and dACC were created by building a box of size 20 x 16 x 16 mm around peak voxels obtained in a meta-analysis (with the x coordinate set to 0 for the dACC) (left dlPFC: MNIxyz = -36, 44, 22, right dlPFC: MNIxyz = 34, 44, 32, dACC: MNIxyz = 0, 18, 42, Fullana et al., 2016). As previously reported (Lonsdorf, Haaker, and Kalisch, 2014), the cortical vmPFC was created by using a box of size 20 x 16 x 16 mm centered on peak coordinates identified in prior studies of fear learning (vmPFC: MNIxyz = 0, 40, -12, e.g., Kalisch et al. (2006), Milad et al. (2007)) with the x coordinate set to 0 to obtain masks symmetric around the midline.”

We have further adjusted Appendix 3-table 3, Figure 3, Table 2 and Figure 6 and have slightly altered the results description to reflect the updated results.

“For BOLD fMRI, both ICC-types suggest rather limited reliability for CS discrimination during acquisition (both ICC_abs_ and ICC_con_ = 0.17) and extinction training (both ICC_abs_ and ICC_con_ = 0.01). For individual ROIs (anterior insula, amygdala, hippocampus, caudate nucleus, putamen, pallidum, nucleus accumbens, thalamus, dACC, dlPFC and vmPFC), ICCs were even lower (all ICCs ≤ 0.001; for full results see Appendix 3-table 3).”

“In contrast to what was observed for SCRs, within-subject similarity was significantly higher than between-subject similarity in the whole brain (*p* <.001) and most of the ROIs for fear acquisition training (see Figure 3A and Appendix 4-table 1). This suggests that while absolute values for similarity might be low, individual brain activation patterns during fear acquisition training at T0 were – at large – still more similar to the same subject’s activation pattern at T1 than to any others at T1. For extinction training, however, no significant differences between within- and between-subject similarity were found for any ROI or the whole brain (all *p*’s >.306; see Figure 3B and Appendix 4-table 1).”

“In stark contrast to the low overlap of individual-level activation (see Table 2A), the overlap at the group level was rather high with 62.00 % for the whole brain and up to 89.80 % for ROIs (i.e., dACC and dlPFC; Jaccard) for CS discrimination during acquisition training (see Table 2B). Similar to what was observed for overlap at the individual level, a much lower overlap for extinction training as compared to acquisition training was observed for the whole brain (5.70 % overlap) and all ROIs (all close to zero).”

“In short, all but one association (CS discrimnation in the NAcc) was positive, showing that higher BOLD response during acquisition was associated with higher BOLD responding during extinction training (see Figure 6). However, the standardized β coefficients are mostly below or around 0.3 except for CS+ associations in the dACC, indicating non-substantial associations for all ROIs and CS specifications that were near absent for CS discrimination. Analysis of CS+ and CS- data was included here as the analysis is based on β maps and not T-maps (as in previous analyses) where a contrast against baseline is not optimal.”

8. Reliability and the Need for Nuancea. The authors structure the manuscript around the premise that reliability is essential in conducting solid individual-differences science, which I agree with wholeheartedly. However, I think the authors rely on relatively arbitrary cut-offs for classifying reliability as good/poor/etc to an extent that is not warranted, particularly in the context of the Discussion, and it takes away from the impact of this effort. As the authors point out, these categorical cut-offs are more guidelines than strict rules, yet the manuscript is structured around the premise that individual-level reliability is problematically poor. Many cut-off recommendations are based on psychometric work on trait self-report measures that usually assume fewer determinants/sources of error than would be seen in neuroscience experiments, which in turn allows for larger ceilings for effect sizes and reliability. The current manuscript does not address this issue and what meaningful (as opposed to good) fear conditioning reliability is when moving away from the categorical cut-offs. In other words, is it possible that the authors actually observed "good" reliability in the context of fear conditioning work, and that this reliability is lower than other types of paradigms is just inherent to the construct being studied?b. The framing of the manuscript could be adjusted, such that less emphasis is placed on arbitrary cut-off metrics and more about what these reliabilities mean in the context of fear conditioning paradigms. To be clear, I think the authors already address this to a degree in their Discussion, but perhaps need to go a step further and expand on the challenges of establishing reliability in this field and explicitly address why common cut-offs are perhaps not appropriate.c. It may be more appropriate to use numbers rather than labels/benchmarks in the reporting of results in the Results section, i.e. reporting the r-value instead of "poor to questionable" etc.

We agree with the reviewer/editor that a focus on the categorical and admittedly arbitrary cut offs may be misleading. We have replaced these (i.e., poor, moderate, high…) with numerical values for reliability in the Results section and explicitly pointed out that these are benchmarks that were developed in a different context and should not be overinterpreted (see figure captions). Changes have been made throughout the manuscript (some examples included below) and a new paragraph was added to the discussion in which we address the points raised by the reviewer/editor:

Abstract:

“While longitudinal reliability was rather limited at the individual level, it was comparably higher for acquisition but not extinction at the group-level.”

Example from figure caption:

“Internal consistency is in the literature often interpreted using benchmarks (Kline, 2013) for unacceptable (< 0.5), poor (> 0.5 but < 0.6), questionable (> 0.6 but < 0.7), acceptable (> 0.7 but < 0.8), good (> 0.8 but < 0.9) and excellent (≥0.9). Common benchmarks in the literature for ICCs are poor (< 0.5), moderate (> 0.5 but < 0.75), good (> 0.75 but < 0.9) and excellent (≥0.9) (Koo and Li, 2016). These benchmarks are included here to provide a frame of reference but we point out that these benchmarks are arbitrary and most importantly derived from psychometric work on trait self-report measures and should hence not be overinterpreted in the context of responding in experimental paradigms in which more sources of potential error are at play (Parsons, 2020).”

Example from results:

“Internal consistency at T0 (see Figure 1A) and T1 (see Figure 1B) of raw SCRs to the CS+ and CS- ranged from 0.54 – 0.85 and for raw SCRs to the US from 0.91 – 0.94 for all phases. In comparison, internal consistency was lower for CS discrimination with values ranging from -0.01 – 0.60.”

Example from discussion:

“Yet, we would like to point out that the values we report may in fact point towards good and not limited longitudinal individual-level reliability as our interpretation is guided by benchmarks that were not developed for experimental data but from psychometric work on trait self-report measures. We acknowledge that the upper bound of maximally overvarable reliability may differ between both use cases as more sources of error are at play in experimental neuroscientific work. The problem remains that predictions in fear conditioning paradigms do not seem feasible for a longer period of time (~ 6 months) given the measures we used here. Thus, a key contribution of our work is that it empirically highlights the need to pay more attention to measurement properties in translational research in general and fear conditioning research specifically (e.g., implement reliability calculations routinely in future studies). To date, it remains an open question what “good reliability” in experimental neuroscientific work actually means (Parsons et al., 2019).”

9. Clarify Implications and Recommendations. The concrete implications of the research, and recommendations arising from it, should be clearly spelled out in the Abstract and Discussion, for the greatest utility.

Please see our answer to comment 4b.